# Selective CK1α degraders exert antiproliferative activity against a broad range of human cancer cell lines

Gisele Nishiguchi[1,9], Lauren G. Mascibroda[2,9], Sarah M. Young[1,9], Elizabeth A. Caine[3], Sherif Abdelhamed[2], Jeffrey J. Kooijman[4], Darcie J. Miller[5], Sourav Das[1], Kevin McGowan[1], Anand Mayasundari[1], Zhe Shi[1], Juan M. Barajas[2], Ryan Hiltenbrand[2], Anup Aggarwal[1], Yunchao Chang[2], Vibhor Mishra[2], Shilpa Narina[6], Melvin Thomas[2], Allister J. Loughran[6], Ravi Kalathur[5], Kaiwen Yu[7], Suiping Zhou[7], Xusheng Wang[7], Anthony A. High[7], Junmin Peng[5,7], Shondra M. Pruett-Miller[6,8], Danette L. Daniels[3], Marjeta Urh[3], Anang A. Shelat[1], Charles G. Mulligan[2], Kristin M. Riching[3], Guido J. R. Zaman[4], Marcus Fischer[1] ✉, Jeffery M. Klco[2] ✉ & Zoran Rankovic[1] ✉

Molecular-glue degraders are small molecules that induce a specific interaction between an E3 ligase and a target protein, resulting in the target proteolysis. The discovery of molecular glue degraders currently relies mostly on screening approaches. Here, we describe screening of a library of cereblon (CRBN) ligands against a panel of patient-derived cancer cell lines, leading to the discovery of SJ7095, a potent degrader of CK1α, IKZF1 and IKZF3 proteins. Through a structure-informed exploration of structure activity relationship (SAR) around this small molecule we develop SJ3149, a selective and potent degrader of CK1α protein in vitro and in vivo. The structure of SJ3149 co-crystalized in complex with CK1α + CRBN + DDB1 provides a rationale for the improved degradation properties of this compound. In a panel of 115 cancer cell lines SJ3149 displays a broad antiproliferative activity profile, which shows statistically significant correlation with MDM2 inhibitor Nutlin-3a. These findings suggest potential utility of selective CK1α degraders for treatment of hematological cancers and solid tumors.

The immunomodulatory imide drugs (IMiDs) thalidomide, lenalidomide, and pomalidomide are the first drugs found to exert their pharmacological effect by inducing protein degradation. IMiDs bind CRBN[1], a substrate recognition domain of E3 ubiquitin ligase CRL4[CRBN]. Altering its specificity results in the recruitment, ubiquitination, and subsequent degradation of proteins not normally targeted by the E3 ligase, termed neosubstrates. Two lymphoid transcription factors, IKZF1 and IKZF3, were first identified as IMiD neosubstrates, rationalizing the clinical efficacy of these drugs in multiple myeloma[2,3].

Subsequently, IMiDs were found to influence the abundance of numerous proteins, including SALL4[4], ZBTB16[5], p63[6], ZFP91[7], ZNF827, RAB28, and RNF166[8,9].

Interestingly, lenalidomide is the only IMiD that was found to induce degradation of casein kinase 1A1 (CK1α), albeit with only a modest efficacy producing incomplete degradation at concentrations of up to 10 μM[10]. Nevertheless, the degradation of CK1α provides a mechanistic basis for lenalidomide's unique clinical efficacy in myelodysplastic syndrome (MDS) patients with deletion of chromosome

5q (del(5q))[11]. This is attributed to the haploinsufficiency of the CK1α gene (*CSNK1A1*) encoded at chromosome 5q32, which heightens MDS sensitivity to the effects of lenalidomide-induced CK1α degradation. CK1α is a ubiquitously expressed cytosolic serine/threonine kinase involved in the regulation of Wnt/β-catenin and p53 signaling[12]. The heterozygous loss of CK1α in del(5q) MDS stabilizes β-catenin, drives self-renewal and dominance of the del(5q) clone, whereas homozygous loss of CK1α, as occurs with lenalidomide-induced degradation of the haplodeficient CK1α, results in p53 induction and selective clonal arrest by virtue of synthetic lethality[13]. It is also interesting that CK1α silencing, using siRNA against *CSNK1A1*, potentiates apoptosis and growth arrest induced by lenalidomide in H929 cells[14], suggesting that a more effective CK1α degrader may display an even greater potency.

Even though del(5q) is also a recurring abnormality in acute myeloid leukemia (AML), lenalidomide has shown limited clinical effect in AML[15]. This was proposed to be due to synthetic lethality that occurs only in del(5q) MDS with haploinsufficient *CSNK1A1*, which is more likely to fail in AML since these leukemias with chromosome 5q abnormalities typically occur with other cytogenetic changes and often carry *TP53* mutations[16]. Nevertheless, CK1α is essential for AML cell survival in vitro and in vivo[17]. Pharmacologic inhibition of CK1α using D4476, a pan-CK1 inhibitor, or CK1α knockdown via lentivirus-mediated shRNA, suppressed proliferation and clone formation by enhancing autophagic flux and apoptosis in both AML cell lines and patient blast cells[18]. In addition, AML patients have higher expression of *CSNK1A1* mRNA than healthy donors, and patients with high *CSNK1A1* have shorter overall survival[18]. Interestingly, a study in a panel of myeloid cancer cell lines showed that lenalidomide promoted the greatest degradation of CK1α in the most sensitive lines, such as HNT-34 and MDS-L, but did not degrade CK1α in the most insensitive line, MOLM-13[19]. In addition to MDS and AML[20], CK1α was also found to promote survival of lymphoma cells[21], as well as a variety of solid tumors such as in lung[22], renal[23], and colorectal[24] cancers, suggesting a potential broader clinical application of compounds that influence CK1α levels or activity. Recently, an elegant medicinal chemistry effort that produced selective CK1α, and dual CK1α/IKZF2 degraders was reported by the Gray and Woo labs, respectively[25,26].

In this work, we employ our proprietary molecular glue library consisting of CRBN binders[27] to probe protein degradation mechanisms via small molecule degraders and neosubstrates. Here we report the results from screening this library against a panel of cancer cell lines and structure-guided SAR exploration that led to the discovery of additional, highly potent, and selective CK1α degraders (Fig. 1).

## Results

### Phenotypic screening of a Molecular Glue Library against a panel of cancer cell lines identifies a hit with a unique cellular profile

We screened our molecular glue library (3630 compounds) against a panel of 9 pediatric cancer cell lines to identify molecules displaying cell-selective sensitivities. In addition to the four cell lines we previously described (MB002, MB004, HD-MB03, and MHHCALL-4)[27], we included five additional AML cell lines (MOLM-13, TF-1, HEL, OCI-AML3, and AML193) to cover a broader range of genetic diversity and cancer vulnerabilities. The library screening was performed in high throughput (384-well plates) by incubating the cells with compounds for 72 h and assessing cell viability via CellTiter-Glo (CTG) in a dose–response manner, as reported previously[27].

As shown by the heatmap in Fig. 1a lenalidomide and pomalidomide were inactive (blue box), indicating that degradation of classical IMiD neosubstrates is not essential for the viability of these cell lines. CC-885[28] and our previously disclosed GSPT degrader (SJ6986)[27,29] were cytotoxic to all cell lines in the panel (yellow box, Fig. 1a), consistent with the previously observed anti-proliferative effect of degrading this essential protein. Clustering of hits by their activity profile across the cancer cell line panel identified one chemical series as cytotoxic and highly selective for MOLM-13 cells with IC$_{50}$ < 1 μM (red box, Fig. 1a). Due to a distinct cell selectivity profile and high toxicity against MOLM-13 cells, we selected compound SJ7095 for further profiling (Fig. 1b).

### Hit profiling identifies CK1α, IKZF1 and IKZF3 as main targets of SJ7095

Having identified SJ7095 as a compound with a potent antiproliferative effect in MOLM-13 cells, we next examined whether the observed effect was CRBN-dependent. In ligand competition experiments, we treated MOLM-13 cells with SJ7095 in dose–response in the presence of high concentrations of lenalidomide (40 μM). Excess lenalidomide

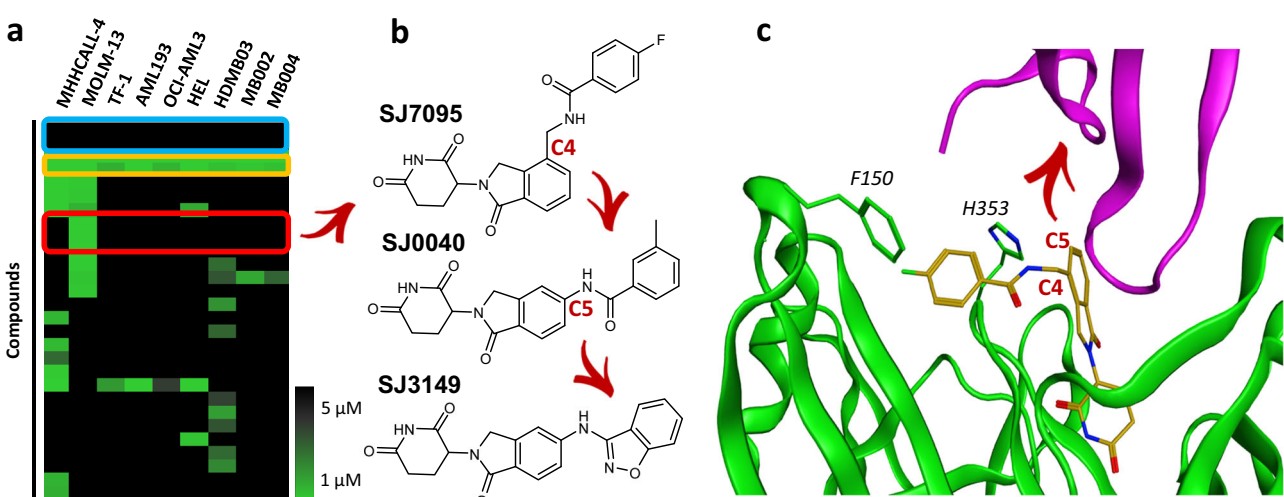

**Fig. 1 | Hit identification and optimization strategy. a** Heatmap of selected screening hits across a panel of acute leukemia (AL) and medulloblastoma (MB) cell lines (cell viability IC$_{50}$ values determined by CTG assays). The blue box highlights lenalidomide and pomalidomide, the yellow box highlights known GSPT degraders CC-885 and SJ6986, red box highlights MOLM-13 hits. **b** Optimization trajectory and chemical structures of the hit and leads. **c** SJ7095 (shown as amber sticks) modeled into the lenalidomide + CRBN (green) + CK1α (purple) complex (PDB: 5FQD)[36]. The red arrow points to the C5-substitution vector that underlines the optimization strategy.

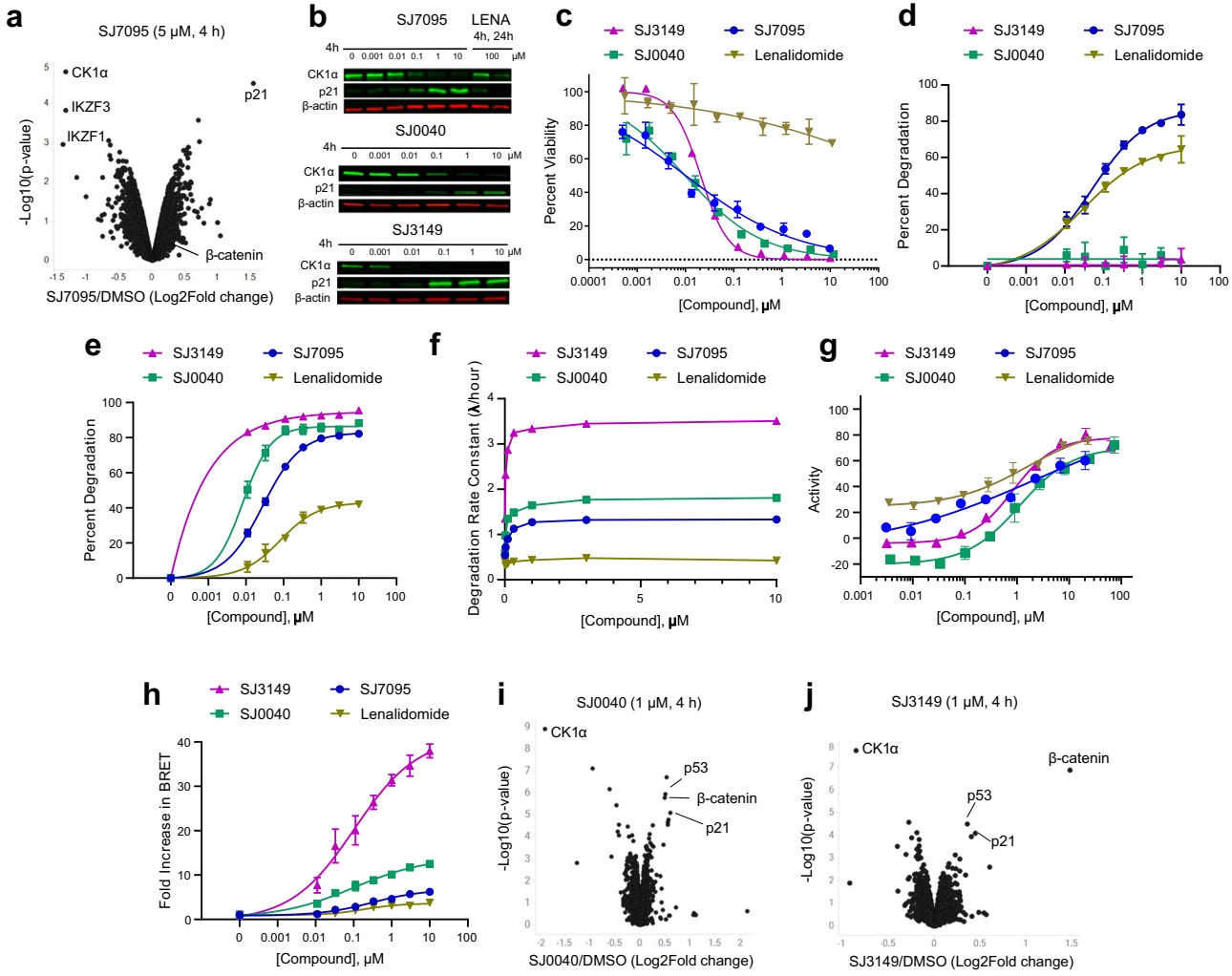

**Fig. 2 | Degradation profiles of hit and leads. a** TMT-Proteomics in MOLM-13 cells following 4 h treatment with SJ7095 at 5 μM concentration. **b** CK1α and p21 protein levels in MOLM-13 cells determined by immunoblotting following treatment with increasing concentrations of test compounds over 4 h, and lenalidomide at 100 μM over 4 and 24 h. The CK1α $DC_{50}$ values were calculated using quantified band intensities from the immunoblotting analysis. Data represents the average from three independent experiments. **c** Viability of MOLM-13 cells measured in the CTG assay after 72 h incubation with rising concentrations of the tested compound. **d** IKZF1 degradation maximum ($D_{max}$) as determined by IKZF1 HiBiT assay. **e** CK1α degradation maximum ($D_{max}$) as determined by CK1α HiBiT assay. **f** CK1α degradation rate as determined by CK1α HiBiT assay. **g** CRBN-binding affinity determined in the fluorescence polarization (FP) displacement assay. **h** Ternary complex formation measured in CK1α-CRBN NanoBRET assay. TMT-Proteomics in MOLM-13 cells following 4 h treatment with 1 μM concentration of: **i** SJ0040 and **j** SJ3149. Data represents the average of at least three independent determinations. Error bars represent the standard error of the mean. Statistical analysis for **a**, **i**, **j**: two-tailed and unpaired *t*-test. Data in Fig. 2 are summarized in Supplementary Table 1.

saturates the CRBN-binding site making it inaccessible for other CRBN modulators, thereby abrogating the antiproliferative effects of SJ7095, as shown in Supplementary Fig. 1a. In an orthogonal approach, we tested SJ7095 in wild-type and CRBN-knockout MOLM-13 cells in a cell viability assay (Supplementary Fig. 1b). Our results unambiguously showed that the antiproliferative activity of SJ7095 was CRBN dependent.

We next carried out a tandem mass tag (TMT) quantitative proteomics experiment[30] to identify the neosubstrates whose degradation is induced by SJ7095. Upon treatment of MOLM-13 cells with SJ7095 over 4 h at 5 μM concentration, significant degradation of only three proteins out of over 9000 identified proteins was observed: CK1α, IKZF1, and IKZF3 (Fig. 2a). This result was consistent at different concentrations and longer timepoint providing confidence in the overall selectivity profile (Supplementary Fig. 2). We confirmed the proteomics results by immunoblot analysis of CK1α in MOLM-13 cells following treatment with increasing concentrations of SJ7095 over 4 h (Fig. 2b). Under these conditions compound SJ7095 induced

dose-dependent reduction of CK1α protein levels with $DC_{50}$ value of 15 nM. The degradation of CK1α was accompanied by a dose-dependent upregulation of p21, which was in agreement with our proteomics data and previously reported studies in which CK1α inactivation resulted in p21-dependent cytotoxicity (Fig. 2b)[31].

The analysis of the publicly available dependency map data (DepMap) showed that MOLM-13 cells were highly dependent on CK1α while less sensitive to IKZF1 and IKZF3 CRISPR knock-out (https://depmap.org/portal). Similarly, MOLM-13 cells were the most dependent of all DepMap AML cell lines on siRNA-mediated silencing of CK1α. Importantly, potent IKZF1 degraders such as classical IMiDs (Fig. 1a), and more recently developed iberdomide (CC-220)[32] showed very little effect on the MOLM-13 cells' viability (Supplementary Fig. 3a). Based on these observations we hypothesized that CK1α degradation was the main driver of the cell viability phenotype produced by compound SJ7095. To further validate this hypothesis, we performed CRISPR genomic editing of MOLM-13 using RNA guides against *CSNK1A1* and measured the relative change of out-of-frame

INDEL frequency at different time points. This provided further evidence of this cell line's dependency on CK1α (Supplementary Fig. 3b). We further confirmed these results by shRNA-targeted silencing of *CSNK1A1* with a GFP reporter and measured the loss of GFP over time corresponding to the proliferative disadvantage of cells with CK1α knock-down (Supplementary Fig. 3c, d). Collectively, these data demonstrated a strong dependency of MOLM-13 cells on CK1α, which rationalized the observed viability effect of compound SJ7095 on this cell line.

Notably lenalidomide, the first and most characterized CK1α degrader, did not show appreciable cytotoxicity in MOLM-13 even at high concentrations (Fig. 2c). Whereas treatment of MOLM-13 cells with 100 nM of SJ7095 over 4 h induced a near complete degradation of CK1α, lenalidomide required a 1000 times higher concentration (100 μM) and longer incubation time (24 h) to achieve a similar CK1α degradation effect (Fig. 2b). This suggests that the observed difference in cytotoxicity stems from lenalidomide's weaker CK1α degradation.

## Structure-guided SAR exploration leads to the discovery of selective CK1α degraders

While the potent and mixed CK1α and IKZF1/3 degradation profile displayed by SJ7095 has potential translational value, we were also interested in developing selective degraders that would be suitable for studying CK1α biology, as well as exploring potential therapeutic approaches for cancers where IKZF1/3 degradation is unnecessary or undesirable[33–35].

To help with understanding the binding mode of SJ7095 and inform our optimization strategy, we performed molecular dynamics studies based on the previously reported crystal structure of lenalidomide in complex with CRBN/DDB1 and CK1α[36]. In the CRBN/DDB1-CK1α model, the distal aromatic group in SJ7095 was aligned with the CRBN surface, and engaged with F150 in a loop (E147-I154) that was partly unresolved in the CRBN-lenalidomide-CK1α complex (PDB: 5FQD)[36], as shown in Fig. 1c. We postulated that this additional interaction with F150 resulted in increased CRBN stability and consequently greater stability of the ternary complex, ultimately leading to greater IKZF1 and CK1α degradation potency when compared to lenalidomide[37]. Importantly, besides the lenalidomide-like interaction between C7 of the solvent-exposed phthalimide ring and G40 of the β-hairpin loop (degron motif) in the N-lobe of CK1α, compound SJ7095 forms contacts only with CRBN. We hypothesized that molecules designed to establish additional stabilizing or destabilizing contacts with neosubstrates may display different degradation selectivity profiles. Specifically, we prioritized substitutions of C5 to engage residues in the β-hairpin loop of CK1α without disturbing the critical C7 interaction with G40 (Fig. 1c).

To evaluate this hypothesis, an array of C5-substituted amid analogs was synthesized and tested in MOLM-13 cells[38]. The most potent analog was 3-methyl-benzamide SJ0040 (Fig. 1b), which inhibited MOLM-13 cell growth with $IC_{50}$ of 14 nM, and reduced CK1α abundance with $DC_{50}$ value of 11 nM and $D_{max}$ 88% (Fig. 2b, c). Another round of optimization, which focused on replacing the benzamide group in SJ0040 with an array of heteroaromatic bioisosteres, led to the discovery of 3-amino-benzisoxazole SJ3149 (Fig. 1b). This compound proved to be the most potent CK1α degrader with $DC_{50}$ of 3.7 nM and $D_{max}$ 95% (Fig. 2b). While $IC_{50}$ of SJ3149 (13 nM) was similar to SJ0040 (14 nM), SJ3149 produced a more robust effect on the viability of MOLM-13 cells (Fig. 2c). As expected, neither of the two compounds affected CK1α levels in CRBN$^{-/-}$ MOLM-13 cells, confirming that their mechanism of action is CRBN-dependent (Supplementary Fig. 1c, d). Wild type and CRBN$^{-/-}$ MOLM-13 cells were also treated with 1 μM SJ7095, SJ0040, or SJ3149 for 24 and 72 h to assess treatment-induced apoptosis by annexin-V staining (Supplementary Figs. 4 and 5). Compared to control cells treated with DMSO, all three compounds led to increased cell death after 72 h (Supplementary Fig. 5a). SJ3149

caused the strongest effect with ~80% of cells dead or dying (Supplementary Fig. 5b). Again, there was no effect on the viability of the CRBN$^{-/-}$ cells (Supplementary Fig. 5).

To better understand CK1α degradation properties, selectivity, and mechanism of action, SJ0040 and SJ3149 were selected for evaluation against lenalidomide and the screening hit, SJ7095, in a panel of biochemical, biophysical, and cellular assays[38]. To measure cellular protein levels, we used cell lines that have been engineered by CRISPR/Cas9 to express endogenous CK1α (HEK293 cells) or IKZF1 (Jurkat cells) with a HiBiT tag, which when coupled with LgBiT expression forms NanoBiT, a bioluminescent protein that can be used for kinetic studies of protein degradation[39]. Compounds were evaluated in a time-dependent and dose–response manner continuously over 24 h period and degradation parameters were calculated as previously reported[39]. Cell viability was also monitored with CellTiter Glo 2.0 (Promega) at the completion of the experiment to ensure that loss of protein was not a consequence of a loss of cell viability. Data collected from these assays were used to generate an extensive multiparameter profile, which included neosubstrate specificity, $DC_{50}$ (half-maximal degradation concentration at 4 h post-treatment), degradation rate constant, $\lambda$ (computed from an exponential decay model fit of the kinetic degradation traces at 10 μM concentration), and $D_{max}$ (maximum protein degradation over the 24 h timeframe at 10 μM concentration).

We first tested the compounds in the IKZF1 HiBiT (Jurkat cells) assay. As expected, lenalidomide and the original hit, SJ7095, induced significant IKZF1 protein degradation, with $DC_{50}$ values of 0.033 and 0.069 μM at 4 h, respectively. Importantly, neither of the two lead compounds, SJ0040 nor SJ3149, had an appreciable effect on the IKZF1 protein abundance ($D_{max} < 10\%$), validating our optimization strategy (Fig. 2d). Furthermore, in the CK1α HiBit assay our degraders demonstrated striking improvements in overall depth of CK1α degradation potency, and rate when compared to lenalidomide, and these improvements were further enhanced with each subsequent lead compound (Fig. 2e, f). In agreement with the immunoblotting results, maximum CK1α degradation induced by lenalidomide was only 42% in the CK1α HiBiT HEK293 cells at the highest concentration (10 μM). In contrast, SJ7095, SJ0040, and SJ3149 induced near-complete degradation of the CK1α protein (Fig. 2e), with $DC_{50}$ values of 35, 10, and 6 nM at 4 h, respectively (Supplementary Table 1). In addition, all three compounds exhibited a dramatic increase in degradation rate (Fig. 2f), with SJ3149 nearly 2-fold higher than SJ0040 and almost 10-fold over lenalidomide, likely driving the observed higher $D_{max}$ and more potent degradation.

Interestingly, despite their superior degradation properties, the three compounds showed up to 8-fold lower affinity to CRBN than lenalidomide in our CRBN fluorescence polarization (FP) displacement assay (Fig. 2g). Indeed, we have previously noted that CRBN affinity is a poor predictor of degradation potency[27]. Experimental evidence reported over recent years suggests that ternary complex stability is a better predictor of degradation efficiency[40]. To determine whether CK1α degradation increased due to the compounds' improved ability to form ternary complex with CK1α, we transfected the CK1α HiBiT cells with a HaloTag-CRBN fusion and measured ternary complex formation by NanoBRET. Following 2 h' treatment, we observed a dose-dependent increase in NanoBRET signal, indicating the formation of a ternary complex with CRBN induced by all our compounds (Fig. 2h). Notably, the ranking in ternary complex formation strongly mirrored the trends observed in degradation rate, $D_{max}$ and potency, with the most robust complex formation induced by SJ3149.

Next, TMT-proteomics studies were performed to determine broader degradation selectivity profiles of our two lead compounds. This experiment showed that after a 4-h incubation at 1 μM concentration both SJ0040 and SJ3149 significantly reduced the abundance of only CK1α out of >9000 proteins identified in MOLM-13 cells (Fig. 2i, j). As expected, increased protein levels of p53, p21, and

β-catenin were detected in the treated cells[31]. To investigate the relationship between β-catenin stabilization relative to CK1α degradation, kinetic monitoring of endogenously tagged HiBiT-β-catenin was performed in live cells and compared to our measurements of CK1α degradation kinetics. Treatment of cells with the lead compounds from each series led to dose-dependent stabilization of β-catenin levels with slower kinetics compared to the observed rapid degradation of CK1α. By calculating the time of onset of β-catenin stabilization for each compound, we found a strong correlation between faster CK1α degradation and shorter β-catenin stabilization time of onset (Supplementary Fig. 6a). Overlaying SJ3149 induced β-catenin stabilization with CK1α degradation kinetics (Supplementary Fig. 6b) revealed an apparent lag in the onset of β-catenin stabilization, which began to show a detectable increase within the first 30 min when CK1α degradation was already >50% at most concentrations. Similar results were observed for p21 induction by immunoblotting in which we observed a near complete loss of CK1α protein by 1 h of SJ3149 treatment yet the induction of p21 was not present until 4 h (Supplementary Fig. 6c). These results support the direct involvement of CK1α in regulation of β-catenin and p21 levels.

We further tested the specificity of SJ3149 considering that previously reported CK1α degraders also lead to degradation of IKZF2[26]. Even though there was a negligible change in IKZF2 protein level in our proteomics studies, we sought to confirm this observation by immunoblotting. In a similar experiment to those shown in Fig. 2b, MOLM-13 cells were treated with SJ3149 for 4 h at increasing concentrations ranging from 1 nM to 10 μM and analyzed by immunoblotting to quantify IKZF2 and CK1α proteins. While at 4 nM concentration SJ3149 induced 50% degradation of CK1α protein, IKZF2 levels were reduced by only ~40% at the highest dose (10 μM), confirming the selectivity of SJ3149 for CK1α over IKZF2 (Supplementary Fig. 6d). These collective data support the claim that CK1α degradation is responsible for the observed cellular phenotypes induced by SJ3149 treatment.

To evaluate the contribution of the CK1α degradation in SJ3149-mediated cytotoxicity, we utilized the degradation-resistant mutant CK1α G40N[41]. MOLM-13 cells were transduced with lentivirus to over-express FLAG-tagged wild type CK1α or the G40N mutant, which removes the essential glycine in the CK1α degron G-loop. Confirmation of protein expression and resistance of the G40N mutant protein to SJ3149 mediated degradation was confirmed by immunoblot (Supplementary Fig. 7a). The transduced cells were then treated with SJ3149 in a dose–response series in triplicate, and cell viability was measured by CellTiter Glo after 72 h. The cells expressing the G40N mutant had a reduced response to SJ3149 compared to cells with only wild type CK1α (Supplementary Fig. 7b), confirming that the G40N mutation in CK1α blocks the effects of SJ3149. A similar observation in the same cell line has been previously reported for a dual CK1α/IKZF2 degrader[26].

## SJ3149 shows a broad antiproliferative profile

After establishing the potency and selectivity of our lead CK1α degraders, we investigated their effect on a panel of AML and ALL cell lines (Fig. 3a). While both compounds displayed cytotoxicity against all eight cell lines (CTG assay), SJ3149 proved to be most potent with IC$_{50}$ values in a single digit nanomolar range, and even sub-nanomolar activity against UCSD-AML1 and PER117 cells (Fig. 3a). Together these results demonstrate that SJ3149 is an exquisitely potent and selective CK1α degrader with a high activity across a range of acute leukemia (AL) cell lines.

To evaluate the compound's effect and potential therapeutic application across a diverse range of human cancers, we profiled SJ3149 in Oncolines® panel consisting of 115 human cancer cell lines, of which 86 were solid tumor and 29 hematologic cell lines, including MOLM-13. Effect on cell viability was monitored using ATPLite™ (Perkin Elmer), which measures intracellular ATP content as an indirect readout of cell number, like CellTiter Glo. IC$_{50}$ and GI$_{50}$ were determined after 72-h treatment of cells with a duplicate 9-point dose range of the compound. SJ3149 potently inhibited the viability of AML cell lines in the panel and cell lines derived from different hematologic neoplasms, such as B-cell and T-cell acute lymphoblastic leukemia (Fig. 3b).

Interestingly, SJ3149 also potently inhibited the viability of multiple cell lines derived from solid tumors, including breast, soft tissue, and tumors of the male and female reproductive system, with several cell lines being inhibited more potently than MOLM-13 (Fig. 3c, Supplementary Data 1). At the other end of the spectrum, multiple cell lines from diverse tumor tissue origins were insensitive (IC$_{50}$ > 31.6 μM), demonstrating that SJ3149 is highly selective in cancer cell lines.

Since tissue-type alone could not explain the cell line responses to compound SJ3149, we investigated whether we could relate cell line sensitivity to genomic alterations in oncogenes or tumor suppressor genes. The mutation status of 38 frequently altered cancer genes was determined and cell lines were grouped according to either 'having' or 'not having' mutations, large deletions or amplifications, and gene translocations in these 38 genes (Supplementary Data 1)[42]. Analysis of variance (ANOVA) was performed to determine whether there were any significant associations between drug response and genetic features of the cell lines. Alterations in TP53 were identified as the only significant drug response marker, where SJ3149 had, on average, 21-fold higher IC$_{50}$ in TP53-altered cell lines compared to cell lines harboring wild-type TP53 (Fig. 3d). This suggests that SJ3149 requires active wild type p53 signaling to exert its inhibitory effect in cell lines.

Our analysis of cell line sensitivity and the basal expression levels of 19,146 genes revealed expression of the p53-regulated gene SESN1 as a strong marker of cell line sensitivity to SJ3149 (Pearson's $r = -0.44$, $p$-value = $4.0*10^{-6}$) (Fig. 3e)[43]. Expression of TP53 moderately, but significantly ($r = -0.27$; $p$-value = 0.01) correlated with sensitivity to SJ3149 (Fig. 3e). These results further support the finding that active p53 signaling is required for cell line response to SJ3149.

To further investigate the mechanism underlying the cell line selectivity of SJ3149, we compared its IC$_{50}$ fingerprint in the viability assays with those of 120 other anti-cancer agents that have been profiled in 102 of the 115 cell lines. This revealed a significant correlation with Nutlin-3a ($r = 0.51$; $p$-value = $1.0*10^{-7}$) (Fig. 4a). Nutlin-3a induces activation of p53 pathway and apoptosis by inhibiting p53 interaction with MDM2, an E3 ubiquitin-protein ligase that regulates p53 levels in cells[44]. Like SJ3149, Nutlin-3a selectively targeted TP53 wild-type cell lines (Fig. 4b). Although SJ3149 and Nutlin-3a shared a similar cell line targeting, SJ3149 inhibited responsive cell lines with low nanomolar or even sub-nanomolar potency, whereas Nutlin-3a inhibited responsive cell lines with micromolar potency. Interestingly, SJ3149 showed the strongest correlation with TP53 expression of all profiled agents (Fig. 4c).

Despite the significant overlap of the cell panel profiles of compound SJ3149 and Nutlin-3a, the Pearson correlation of 0.51 indicated that there were also notable differences in the cellular context required for optimal cell-killing activity of the two compounds. While having a preference for TP53 wild-type cell lines, SJ3149 also inhibited several TP53-altered cell lines that were insensitive to Nutlin-3a (Fig. 4b). When we further investigated the different positions of TP53 missense mutations, mutation of the lysine at position 132 to either glutamine or arginine in BT-20 or KU812, respectively, conferred high sensitivity to SJ3149. SJ3149 had, on average, 2500 times lower IC$_{50}$ in these two cell lines, compared to cell lines harboring missense mutations on other positions in TP53 (Fig. 4d). Given that lysine at position 132 is a reported ubiquitination site[45], a selective CK1α degrader may therefore be more advantageous in targeting specific TP53 mutant cancers compared to direct inhibition of the p53–MDM2 interaction.

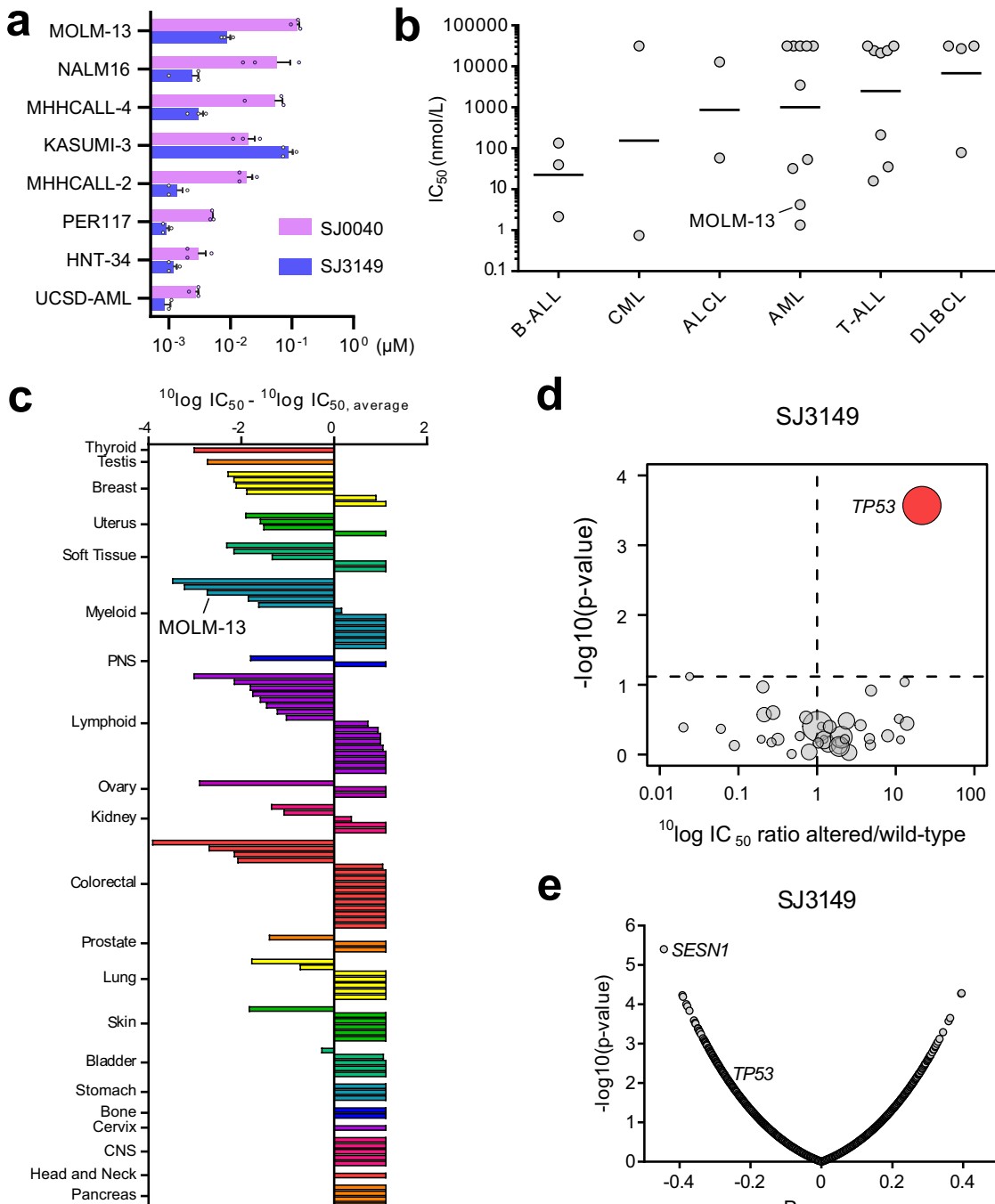

**Fig. 3 | Analysis of SJ3149 activity in a broad range of human cancer cell lines.**
**a** $IC_{50}$ values of SJ0040 and SJ3149 in a panel of AL cell lines. The data are plotted as the mean ± SEM from three independent experiments. **b** Scatterplot of the $IC_{50}$ distribution of SJ3149 in 29 hematologic cell lines. Cell lines were grouped based on their disease subtype. Dots show the mean $IC_{50}$ value (in nM) as derived from duplicate 9-point dilution series for each cell line. Horizontal solid lines indicate the geometric means, as derived from the independent samples of each group (B-ALL $n = 3$, CML $n = 2$, ALCL, $n = 2$, AML $n = 10$, T-ALL $n = 8$, DLBCL $n = 4$). **c** SJ3149 $IC_{50}$ values for the 115 cancer cell lines relative to the panel average $IC_{50}$. A negative value indicates a below-average $IC_{50}$ value. Bars are based on the mean $IC_{50}$ value as derived from duplicate 9-point dilution series for each cell line. Cell lines were grouped and colored based on their tissue of origin. **d** Volcano plot comparing compound SJ3149 $IC_{50}$ differences between altered and wild-type cell lines for 38

established cancer genes. The red node indicates significantly higher $IC_{50}$ in the *TP53*-altered cell lines. **e** Volcano plot of Pearson correlations between SJ3149 $IC_{50}$ values and basal expression levels of 19,146 genes in 99 cell lines. Plots in **d** and **e** were generated using $^{10}$log $IC_{50}$ values (in nM) derived from duplicate 9-point dilution series for each cell line. For results in **d**, the significance of $IC_{50}$ shifts was determined by two-sided Type II ANOVA as implemented by the *'Anova()'* function from the *'car'* package in R. Benjamini–Hochberg multiple testing correction was performed using the *'p.adjust()'* function from the *'stats'* package in R. Adjusted $p$-values < 0.2 were considered significant. For results in **e**, correlations were determined using the *'cor.test()'* function from the *'stats'* package in R, using the Pearson method, pairwise complete observations, and a two-sided alternative hypothesis.

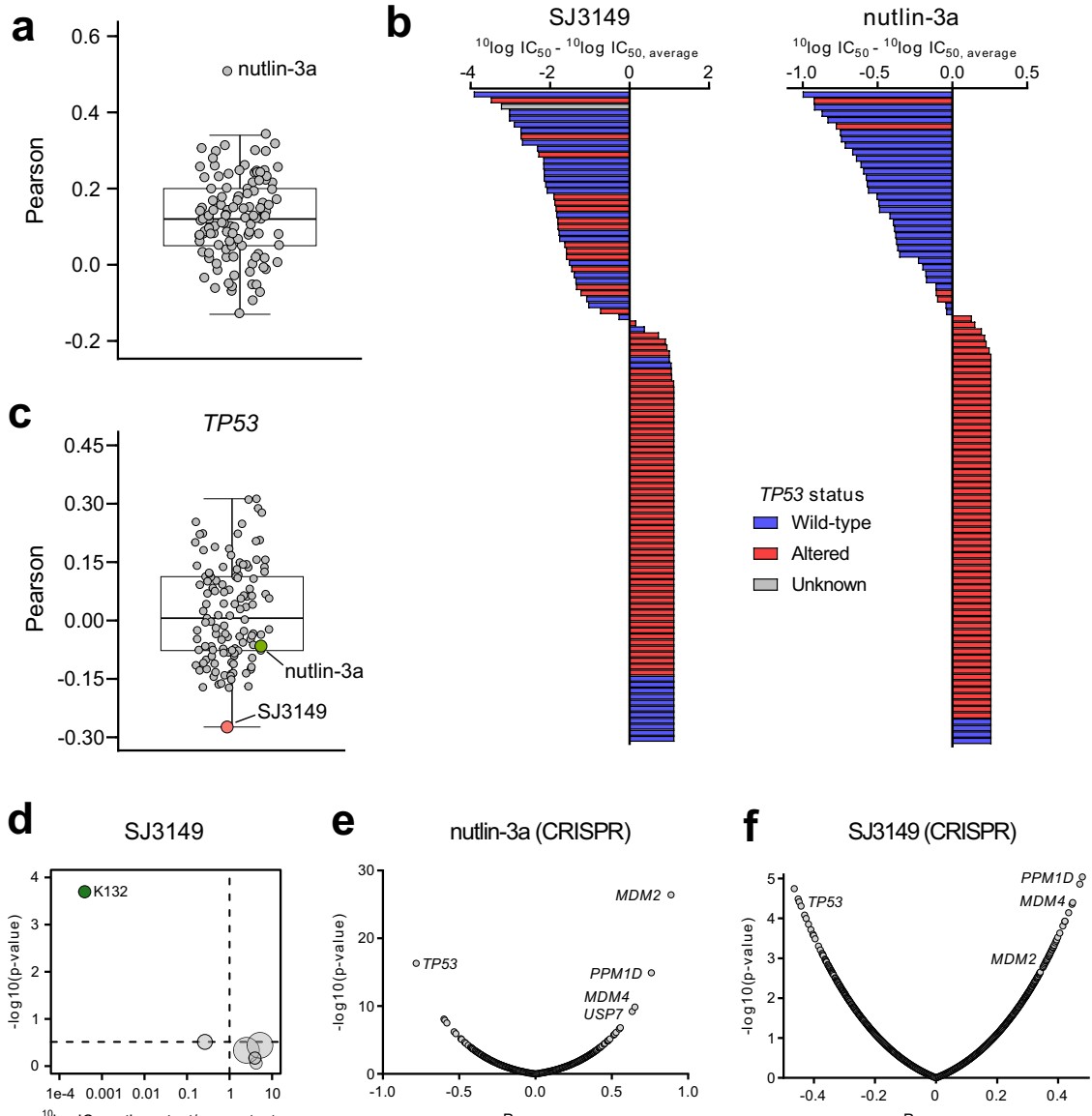

**Fig. 4 | Determinants of cell line response to SJ3149. a** Boxplot of Pearson correlations between SJ3149 IC$_{50}$ values and IC$_{50}$ values of 120 anti-cancer agents in 102 cell lines. **b** Waterfall plots of cellular response to SJ3149 (left) and Nutlin-3a (right). Cell lines were colored based on the genomic status of *TP53*. Bars are based on the mean IC$_{50}$ value as derived from duplicate 9-point dilution series for the 115 (SJ3149) or 102 (Nutlin-3a) cell lines. **c** Boxplot of Pearson correlations between basal expression levels of *TP53*, and IC$_{50}$ values of SJ3149 and the 120 anti-cancer agents. Correlations are based on 99 or 95 cell lines for SJ3149 and the 120 other agents, respectively. **d** Volcano plot comparing the compound SJ3149 IC$_{50}$ differences between cell lines harboring *TP53* missense mutations. The green node indicates significantly lower IC$_{50}$ in cell lines harboring a missense mutation on residue K132. Volcano plot of the Pearson correlations between drug IC$_{50}$ values and CRISPR dependency scores of 17,453 genes in 77 cell lines for **e** Nutlin-3a and

**f** SJ3149. Correlations are based on 78-cell lines. Plots in **a**, **c**–**f** were generated using $^{10}$log IC$_{50}$ values (in nM) derived from duplicate 9-point dilution series for each cell line. For results in **a**, **c**, **e**, and **f**, correlations were determined using the 'cor.test()' function from the '*stats*' package in R, using the Pearson method, pairwise complete observations, and two-sided alternative hypothesis. For results in **d**, the significance of IC$_{50}$ shifts was determined by two-sided Type II ANOVA as implemented by the 'Anova()' function from the '*car*' package in R. Benjamini–Hochberg multiple testing correction was performed using the '*p*.adjust()' function from the 'stats' package in R. Adjusted *p*-values < 0.2 were considered significant. For the boxplots in **a** and **c**, bounds of boxes represent the first and third quartiles, the center indicates the median. Whiskers extend from the upper and lower bounds of the box to the largest or smallest value no further than 1.5 times the interquartile range. Outliers extending beyond these ranges are plotted individually.

To provide further insight into the cellular targeting of SJ3149 and differences from Nutlin-3a, we correlated the IC$_{50}$ profiles of both SJ3149 and Nutlin-3a to cell line dependency profiles of more than 17,000 genes, based on large CRISPR screens[46]. As expected, the IC$_{50}$ profile of Nutlin-3a strongly correlated with dependency on MDM2 and MDM4, while *TP53* was strongly anti-correlated (Fig. 4e). Although SJ3149 was also significantly correlated with dependency on MDM2 and MDM4, these correlations were reduced compared to Nutlin-3a (Fig. 4f). Notably, *CSNK1A1* CRISPR dependency scores

were not significantly correlated with response to SJ3149 (*r* = 0.18; *p*-value = 0.12), while *CSNK1A1* was previously identified as a pan-essential gene in two distinct CRISPR screens[47]. It has been proposed that, for CRISPR pan-essential genes, knockdown using RNAi might better reflect pharmacologic inhibition of the protein target, owing to the partial gene suppression induced by RNAi[48]. Indeed, the correlation between SJ3149 IC$_{50}$ values and *CSNK1A1* RNAi scores exhibited a statistically significant improvement (*r* = 0.26; *p*-value = 0.03).

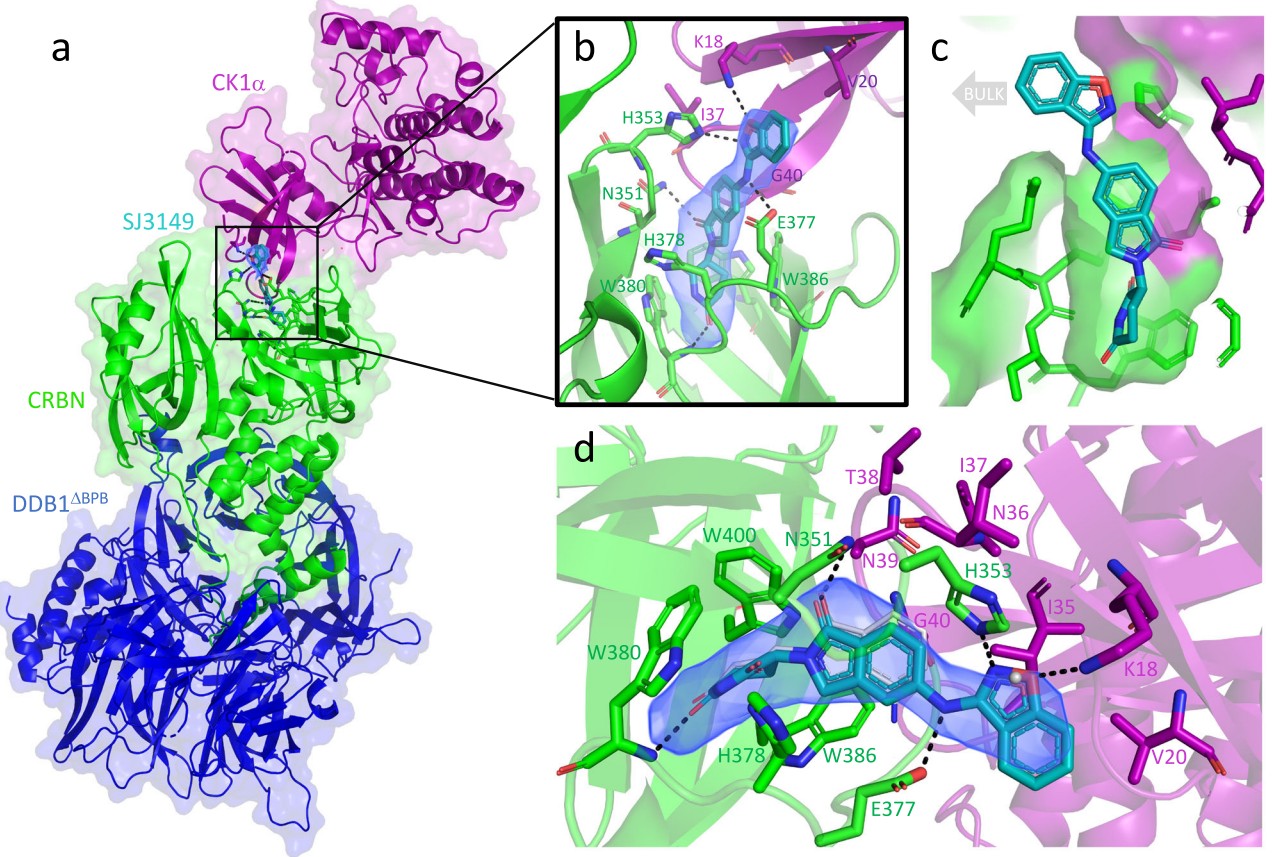

**Fig. 5 | Quaternary complex of CK1α + CRBN^Δ1-40 + DDB1^ΔBPB + SJ3149.**
**a** Quaternary complex of CK1α (purple), CRBN (green), and DDB1 (blue) in the presence of molecular glue SJ3149 (shown as cyan sticks). **b** The magnified region of the binding interface of CK1α and CRBN that accommodates SJ3149 with hydrogen bonds is shown as dashed lines. **c** Surface representation of the SJ3149 binding pocket with the benzisoxazol exposed to bulk solvent and available for chemical modification. **d** The Fo–Fc electron density map (rendered at 2.5 sigma) of the

ligand allows the unambiguous placement of SJ3149 in the binding site with the benzisoxazol moiety extending to directly H-bond with K18 of CK1α, and H353 and E377 of CRBN. Overlay of the lenalidomide quaternary complex (5FQD; lenalidomide shown as thin gray sticks) shows similar accommodation of the shared glutarimide moiety and the displacement of a water molecule (gray sphere) that was observed in the lenalidomide-bound complex upon SJ3149 binding.

The results of the cancer cell line profiling provide a rationale for the application of selective CK1α degraders across a wide range of hematological and solid tumors. The cell line inhibition profile of SJ3149 strongly suggests preferential targeting of cell lines with wild-type *TP53* and certain *TP53* mutations, with marked potency and selectivity differences over Nutlin-3a.

## Structural basis of CK1α + CRBN + DDB1 quaternary complex formation by SJ3149

To understand the molecular basis for the improved degradation potency of SJ3149 compared to lenalidomide, we pursued crystallography studies of the CK1α + CRBN + DDB1 + SJ3149 quaternary complex (Fig. 5a). The resulting 3.45 Å crystal structure shows that the core scaffold which SJ3149 shares with lenalidomide is accommodated similarly in the hydrophobic pocket of CRBN (Fig. 5b, c), including direct interaction with residue E377 (Fig. 5d; and Supplementary Fig. 8a). Interestingly, E377 was previously postulated to contribute to substrate specificity, as it preferentially interacts with lenalidomide over pomalidomide and thalidomide as targets for CK1α[36]. Unlike lenalidomide, which primes the CRBN interface for interaction with CK1α, SJ3149 forms several direct interactions with CK1α (Fig. 5b). SJ3149 engages with the CK1α N-terminal domain (NTD) via a hydrogen bond with K18 and several hydrophobic contacts with V20 and residues of the β-hairpin degron loop (from I35-G40) of CK1α (Fig. 5d), including G40 which is key for CRBN recruitment (Supplementary

Fig. 8b)[8,49]. Mutation of G40 to asparagine in the CK2α isoform prevents binding to CRBN, rendering CK2α undegraded, which exemplifies the importance of G40 in CRBN recruitment[36]. Upon extending into the solvent-exposed space, the benzisoxazol moiety of SJ3149 displaces a water molecule that is present in the lenalidomide-bound complex (PDB: 5FQD)[36] and forms a hydrogen bond with H353 of the CRBN C-terminal domain (CTD) (Fig. 5d). These anchoring hydrophobic and hydrogen bonding interactions at the CRBN–CK1α interface corroborate our modeling prediction that switching from C4 to C5 substitution provides a vector for a closer ternary complex.

It has been previously reported that upon ligand binding, the open apo state of CRBN rearranges into the closed state that is stabilized by a "sensor loop" (residues 351–354), which adopts a β-hairpin conformation upon closure and enables substrate binding and degradation[37]. When superimposed onto open (PDB: 8CVP) and closed (PDB: 8D81) CRBN, our structure confirms that CK1α is bound to the closed state of CRBN (Supplementary Fig. 9). A portion of the N-terminal Lon "belt" domain becomes ordered and wraps around the thalidomide binding domain (TBD) of CRBN for added stabilization (Supplementary Fig. 10). With SJ3149 forming a hydrogen bond with K18 and additional hydrophobic contacts with the CK1α degron-loop (Fig. 5b, d) that is absent in the lenalidomide-bound complex, our structure proposes a molecular basis for SJ3149's increased ternary complex stabilization and consequent CK1α degradation potency compared to lenalidomide.

**Table 1 | Pharmacokinetic parameters in mouse for SJ3149 obtained after intravenous (IV), oral (PO), and intraperitoneal (IP) administration**

| Route | Dose (mg/kg) | $T_{max}$ (hr) | $C_0/C_{max}$ (ng/mL)[a] | $AUC_{last}$ (h*ng/mL) | $T_{1/2}$ (h) | CL (mL/min/kg) | $V_{ss}$ (L/kg) | %F |
|---|---|---|---|---|---|---|---|---|
| IV | 3 | NA | 4510.70 | 1374.61 | 0.77 | 36.24 | 0.67 | NA |
| PO | 50 | 0.25 | 4740.05 | 2848.12 | 2.57 | – | – | 12 |
| IP | 50 | 0.25 | 6683.49 | 16,983.57 | 3.08 | – | – | 74 |

*NA* not applicable.

[a]Back extrapolated concentration at time zero in the IV group.

While our structure revealed several key distinct interactions, the rest of the protein–protein interface appears largely unchanged compared to the lenalidomide complex, despite a subtle but notable rotation of CK1α (Supplementary Movie 1). Finally, to establish the activation state of CK1α in the complex, we superimposed our structure onto the active state of CK1α. Although no ATP was bound to its nucleotide binding site, CK1α's DFG and HRD motifs flanking the disordered activation loop were in the same active state observed in the lenalidomide complex (Supplementary Fig. 11)[36,50].

To rationalize the degradation selectivity of SJ3149 for CK1α over IKZF1, we aligned our structure of SJ3149 with a previously reported structure of IKZF1 in complex with lenalidomide and CRBN + DDB1 (6H0F)[8]. This model suggests that the change of the hydrophobic residue I35 in CK1α to the polar residue Q46 in the equivalent position in IKZF1 may destabilize the aromatic benzisoxazole moiety of SJ3149 and, at least in part, contribute to the compound's degradation selectivity over IKZF1 (Supplementary Fig. 12).

### SJ3149 induces CK1α degradation in vivo

We generated mouse and human in vitro absorption, distribution, metabolism, and excretion (ADME) data for SJ3149 (Supplementary Table 3). Based on a favorable ADME profile, SJ3149 progressed to pharmacokinetics experiments in CD1 female mice (Table 1).

Upon IV administration at 3 mg/kg dose SJ3149 showed a rapid plasma clearance with a terminal elimination half-life of 0.77 h. The observed discrepancy between in vitro microsomal and hepatocyte data with in vivo clearance suggests that the non-P450 mechanism dominates the compound's clearance in vivo. Following a single administration by oral route at 50 mg/kg dose, the compound showed an improved half-life of ~3 h, with peak plasma concentration $C_0/C_{max}$ of 4740.05 ng/mL (12.59 μM) and oral bioavailability of 12%. Upon IP administration of the same dose (50 mg/kg), SJ3149 displayed $C_{max}$ ($C_0/C_{max}$ = 6683.49 ng/mL), exposure ($AUC_{last}$ = 16,983.57 h*ng/mL), and bioavailability (74%) considerably improved over the oral route (Table 1).

Based on the compound's overall PK profile, we decided to proceed to the PD study with the IP route of administration and two dosing regimens: 50 mg/kg once per day and 50 mg/kg twice per day. For the PD experiment, NSG mice engrafted with MOLM-13 cells were treated for 48 h with SJ3149 or DMSO vehicle in 4 groups: vehicle, 50 mg/kg once per day, or vehicle or 50 mg/kg twice per day (Fig. 6). After 48 h, the mice were sacrificed and human cells were isolated from the collected bone marrow and lysed for western blot analysis. Both treatment groups showed significant degradation of CK1α, with 50 mg/kg twice per day producing the lower CK1α protein level (Fig. 6).

### Discussion

Although CK1α is a potential target for hematological indications[20] there are no reported inhibitors nor degraders of this kinase with adequate potency and selectivity to enable pre-clinical validation studies. Lenalidomide is the only FDA-approved drug known to induce, albeit weak, degradation of the CK1α protein. This may explain, at least in part, why lenalidomide shows meaningful clinical effects in MDS

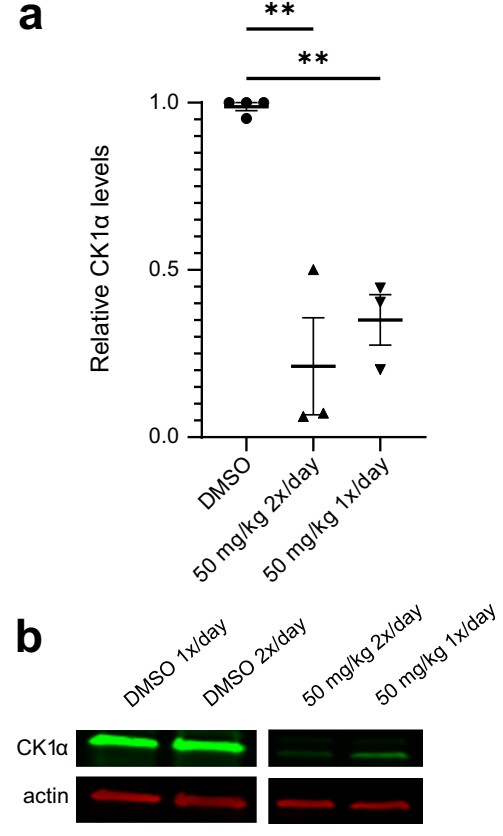

**Fig. 6 | Effect of SJ3149 on CK1α protein levels in vivo.** Data was obtained after indicated IP dosing regimens. **a** Relative CK1α expression calculated by quantified western blots of isolated human cells from the bone marrow of treated mice ($n = 3$ for treatment groups, $n = 2$ for vehicle groups; DMSO vehicle control samples were combined from both groups for graphing; bars represent mean and SEM. One-way ANOVA with multiple comparisons to DMSO: 50 mg/kg 2×/day $p = 0.0023$, 50 mg/kg 1×/day $p = 0.0080$. **b** Representative blot shows one sample from each treatment group (one lane was excluded from analysis and removed from the blot shown here).

characterized by CK1α haploinsufficiency but lacks efficacy against AML in pre-clinical and clinical settings[51]. Herein, we report the discovery of potent and selective CK1α degraders with activity in a broad range of acute leukemia and solid tumor cell lines.

Screening of our library of CRBN-directed molecular glues against a panel of pediatric cancer cell lines led to the identification of SJ7095 with high cytotoxicity against MOLM-13 cells ($IC_{50}$ = 71 nM), and potent CRBN-dependent degradation of CK1α, IKZF1 and IKZF3 proteins. This finding provided us with an initial confirmation that pharmacologically induced CK1α degradation can phenocopy genetic knockdown experiments in MOLM-13 cells. Since our CRISPR and shRNA experiments showed that the loss of CK1α protein alone is sufficient to affect the viability of MOLM-13 cells, we endeavored to develop selective CK1α degraders. Our structure-informed optimization focused on

moving the substitution vector on the lenalidomide core of SJ7095 to establish direct interactions with the degron motif. Resulting SJ0040 and SJ3149 degraders displayed fast and potent CK1α degradation, proteome-wide selectivity, and improved antiproliferative effects in MOLM-13 cells. Both compounds were superior to lenalidomide with respect to the depth, kinetics, and potency of CK1α degradation. These observed improvements can be rationalized by CRBN-CK1α ternary complex formation induced in cells by the compounds. Indeed, the ranking in ternary complex formation directly reflected the trends observed in degradation potency, with the most robust complex formation induced by SJ3149 (Fig. 2h). The quaternary structure of CK1α + CRBN + DDB1 + SJ3149 (Fig. 5) showed that, in contrast to lenalidomide, SJ3149 forms several direct interactions with CK1α, further validating our optimization hypothesis and rationalizing the improved ternary complex formation and ultimately CK1α degradation potency displayed by this compound.

SJ3149 also displayed superior activity against a panel of acute leukemia cell lines, with $IC_{50}$ values in the low nanomolar to sub-nanomolar range. Furthermore, in a larger panel of 115 human cancer cell lines SJ3149 potently decreased the viability of multiple solid tumor cell lines, including breast, soft tissue, and tumors of the male and female reproductive system, with several cell lines showing even greater sensitivity than MOLM-13 (Fig. 3). Interestingly, SJ3149 activity showed a strong correlation with wild-type *TP53* expression. An in-depth analysis of $IC_{50}$ fingerprints produced in this panel by SJ3149 and 120 other anti-cancer agents revealed a significant correlation with MDM2 inhibitor Nutlin-3a (Fig. 3e)[44]. This pharmacological similarity is not surprising, considering the role of CK1α in MDM2 activation and p53 pathway inhibition[31]. Interestingly, SJ3149 also inhibited several *TP53*-altered cell lines, such as BT-20 and KU812, suggesting that CK1α degraders may offer potential therapeutic opportunities for cancers that are insensitive to MDM2 inhibitors. Despite these results, there are several limitations of our phenotypic screening approach that rely solely on the cell viability as the assay readout. One is that degraders of proteins that are not essential for the cells featuring in the screening panel would be missed, while they could still be oncodrivers in cancers not represented in the panel. For example, since none of the cell lines in our panel are sensitive to IKZF1 degradation we would miss discovering IKZF1 degraders, which are a component of the front-line therapies for multiple myeloma. In addition, weaker degraders exerting only a small effect on cell viability may also be missed, even if they could provide an attractive starting point for optimization. Moreover, the degradation of previously reported neosubstrates associated with a strong phenotype, such as GSPT1, can generate unproductive follow-up work.

Overall, our study provides a rationale for the application of selective CK1α degraders across a wide range of hematologic and solid tumors. The cell line inhibition profile of SJ3149 strongly suggests effective targeting of *TP53* wild-type cell lines, with marked potency and selectivity differences over Nutlin-3a. More generally, this study further supports the utility of a library of diverse CRBN binders in lead discovery, and the ability to rationally optimize molecular glue potency and selectivity.

## Methods

### Ethical Statement
The PK study was conducted at the AAALAC-accredited facility of Sai Life Sciences Limited, Pune, India, in accordance with the Study Protocol SAIDMPK/PK-20-08-538. All animals were maintained in standard animal cages under conventional laboratory conditions (12 h/12 h light/dark cycle, 22 °C) with ad libitum access to food and water. All procedures of the present study were performed in accordance with the guidelines provided by the Committee for the Purpose of Control and Supervision of Experiments on Animals (CPCSEA) as published in The Gazette of India, on December 15, 1998. Prior approval from the Institutional Animal Ethics Committee (IAEC) was obtained before the initiation of the study.

For the xenograft transplantation study, female NSG mice (NOD.Cg-*Prkdc^scid^ Il2rg^tm1Wjl^*/SzJ, Strain #005557, The Jackson Laboratory) aged 6–8 weeks were transplanted with 100,000 MOLM-13 cells by tail-vein injection. 10 days post-transplantation, mice were treated with SJ3149 via intraperitoneal injections for the following 2-day dosing schedule: 50 mg/kg (1 dose every 12 h), and 50 mg/kg (1 dose every 24 h) with $n = 3$ per group. Control groups ($n = 3$) were treated with DMSO either 1 dose every 24 h or 1 dose every 12 h with equal volume as treatment groups. Mice were euthanized 48 h after the first treatment and bone marrow cells were collected. Murine cells were depleted using the EasySep Mouse/Human Chimera isolation kit (STEMCELL Technologies #19849). NSG mice were housed in micro-isolation cages and kept under barrier conditions to keep them specific pathogen-free. The lights for NSG mice are on an automated 12-h on, 12-h off light cycle. The mouse room has a separate thermostat and humidistat to control temperature and humidity at the room level. Sex was not considered beyond female mice being most commonly used for cell line-derived xenografts. Ethical approval by the St. Jude Institutional Animal Care and Use Committee.

### Molecular glue library screening
The primary screen was performed in 384-well format using the Cell-Titer Glo (CTG) luminescent cell viability assay (Promega) as previously described[27]. 3630 compounds from the St. Jude proprietary molecular glue library were screened against 9 human cancer cell lines (HD-MB03, MB004, MB002, MHH-CALL4, MOLM-13, TF-1, HEL, OCI-AML3, AML193). Each cell line was cultured in the complete medium recommended by the vendor and seeded in Corning 8804 BC white 384-well assay plates at densities of 1000, 1000, 1500, 7500, 1250, 156, 625, 1250, 1250 cells per well for HD-MB03, MB004, MB002, MHH-CALL4, MOLM-13, TF-1, HEL, OCI-AML, AML193, respectively. After overnight incubation at 37 °C in a humidified 5% $CO_2$ incubator, cells were treated with compounds in dose–response format using a Pintool on a Biomek $FX^P$ Laboratory Automation Workstation (Beckman Coulter). After 72 h of incubation, cell proliferation was assessed using a CellTiter-Glo (CTG) luminescent cell viability assay (Promega) according to the manufacturer's instructions. Luminescence signal was measured using an EnVision plate reader (PerkinElmer).

### Cell lines and cell culture
HD-MB03 cell line was obtained from Drs. Milde, Witt and Deubzer[52]. HD-MB03 and MB004 cells were grown in a neurobasal medium supplemented with glutamine, streptomycin, penicillin, B27, EGF, bFGF, and heparin as previously described[53]. MB002 and MB004 cells were obtained from Dr Yoon-Jae Cho[54]. MB002 cells were cultured in 80% neurobasal medium supplemented with glutamine, streptomycin, penicillin, B27, and 20% conditioned media from previous passages; EGF, bFGF, and heparin were also added to the medium. MHH-CALL4, NALM-16, MHH-CALL-2 cells were obtained from the German Collection of Microorganisms and Cell Cultures GmbH (DSMZ, Germany). PER-117 cells were obtained from Telethon Kids Institute (Perth, Australia). The cells were cultured in RPMI1640 medium supplemented with 10% FBS (Hyclone), Penicillin/Streptomycin (100 units/mL), and Glutamine (100 μM) except for MHH-CALL-2 cells where 20% FBS were used. MOLM-13 cells were obtained from DSMZ (ACC 554) and cultured in RPMI-1640 media (Gibco) supplemented with 20% FBS (R&D Systems) and 100 U/ml penicillin-streptomycin (Gibco). HNT-34 cells were purchased from DSMZ (ACC 600) and cultured in RPMI-1640 media supplemented with 20% FBS and 100 U/ml penicillin-streptomycin. Kasumi-3 cells were purchased from ATCC (CRL-2725) and cultured in RPMI-1640 media supplemented with 20% FBS and 100 U/ml penicillin-streptomycin. UCSD-AML1 cells were purchased from DSMZ (ACC 691) and cultured in RPMI-1640 media supplemented

with 20% FBS, 100 U/ml penicillin–streptomycin, and 10 ng/ml human GM-CSF (PeproTech). TF-1 cells were purchased from DSMZ (ACC 334) and cultured in RPMI-1640 media supplemented with 20% FBS, 100 U/ml penicillin–streptomycin, and 5 ng/ml human GM-CSF. OCI-AML3 cells were purchased from DSMZ (ACC 690) and cultured in Alpha-MEM media (Invitrogen) supplemented with 20% FBS and 100 U/ml penicillin–streptomycin. HEL cells were obtained from Dr. Charles Mulligahn and cultured in RPMI-1640 media supplemented with 10% FBS and 100 U/ml penicillin-streptomycin. AML193 cells were purchased from DSMZ (ACC 549) and cultured in IMDM supplemented with 5% FBS, 100 U/ml penicillin–streptomycin, Insulin–Transferrin–Selenium (ITS-G, Gibco), and 5 ng/ml human GM-CSF. Cell identity was confirmed by STR profiling using PowerPlex® Fusion System (Promega).

### Modeling SJ7095 in CK1α
Molecular modeling work was done using the Schrodinger Maestro molecular modeling package (Schrödinger Release 2019-4: LigPrep, Schrödinger, LLC, New York, NY, 2019). A lenalidomide-bound complex structure was obtained from the PDB (PDB code: 5FQD)[55]. Only chains B (CRBN) and C (CK1α) were retained and the resulting structure was prepared using the Protein Preparation Wizard[56] such that the missing loop Gln148-Glu153 in CRBN was reconstructed. The tool allowed hydrogen-bond optimization and restrained minimization of the complex (converge heavy atoms to RMSD 0.3 Å). This structure was relaxed using a molecular dynamics simulation at 300 K and 1.01325 bar pressure using a water-box. SPC waters and an orthorhombic boundary box with 15 Å buffer were chosen and the system was neutralized by adding Cl- ions. OPLS3e forcefield was used to simulate the system in GPU-accelerated Desmond (Schrödinger Release 2019-4: Desmond Molecular Dynamics System, D. E. Shaw Research, New York, NY, 2019. Maestro-Desmond Interoperability Tools, Schrödinger, New York, NY, 2019). An NPT ensemble was used with a timestep of 2 fs and a Coulombic short-range cutoff radius was set at 9 Å. Upon completion of the simulation, the water molecules were removed and SJ7095 was docked in the place of the Lenalidomide molecule using the Schrodinger Glide Extra Precision (XP) method[57] and the docked complex was further simulated for 500 ns using the parameters defined earlier, to obtain the SJ7095–CRBN–CK1α complex.

### Proteomics in MOLM-13
Each compound was tested in three wells of 2 million MOLM-13 cells each. Cells were treated in tissue culture-treated six-well plates in 3 ml total media volume and incubated at 37 °C for 4 h. Cells were then collected and washed with DPBS. Washed cells were centrifuged at 400×g for 5 min at 4 °C, the supernatant was removed, and pellets were snap frozen in liquid nitrogen and stored at −80 °C until sample submission.

### Protein digestion and peptide isobaric labeling by *-*+TMT reagents
The experiment was performed with a previously optimized protocol[30] with slight modification. Cell pellets were lysed in lysis buffer (50 mM HEPES, pH 8.5, 8 M urea, and 0.5% sodium deoxycholate). To profile the whole proteome, the protein lysates (approximately 100 µg of protein per sample) were proteolyzed with LysC (Wako) at an enzyme-to-substrate ratio of 1:100 (w/w) for 2 h at 21 °C. Following this the samples were diluted to a final 2 M Urea concentration, and further digested with trypsin (Promega) at an enzyme-to-substrate ratio of 1:50 (w/w) for at least 3 h. The peptides were reduced by adding 1 mM DTT for 30 min at 21 °C followed by alkylation with 10 mM iodoacetamide (IAA) for 30 min in the dark. The unreacted IAA was quenched with 30 mM DTT for 30 min. Finally, the digestion was stopped by adding trifluoroacetic acid (TFA) to 1%, desalted using C18 cartridges (Harvard Apparatus), and dried by speedvac. The purified peptides

were resuspended in 50 mM HEPES (pH 8.5), and labeled with TMT reagents (Thermo Scientific). The differentially labeled samples were pooled equally, desalted, and dried for the subsequent peptide fractionation. Peptide analysis by two-dimensional liquid chromatography–tandem mass spectrometry (LC/LC–MS/MS) and MS data analysis are described in Supplementary Material.

### Western blotting
Frozen pellets of ~1 million cells were lysed in 150 µl of SDS lysis buffer (60 mM Tris/HCl pH 7, 10% glycerol, 2% SDS, 5% beta-mercaptoethanol, 0.02% bromophenol blue, 0.5% protease and phosphatase inhibitor cocktail (Sigma Aldrich)). When cells were resuspended, samples were heated for 3 min at 99 °C then put on ice. Samples were then fragmented by sonication for 10 0.5-s pulses (550 Sonic Dismembrator, Fisher Scientific) and heated again at 99 °C for 2 min. 15 µl of each sample was added to the wells of a 4–20% polyacrylamide gel (Bio-Rad). A mixture of 2 µl of Precision Plus Protein Unstained Standards and 2 µl of Precision Plus Protein All Blue Standards (Bio-Rad) was used for the size marker. After electrophoresis, gels were transferred to nitrocellulose membrane using the Bio-Rad Trans-Blot Turbo system on the mixed molecular weight setting. Membranes were blocked for 1 h in Li-Cor Intercept Blocking Buffer then probed overnight at 4 °C with primary antibody, at 1:1000 dilution or 1:2000 dilution for actin, in Li-Cor Intercept Antibody Diluent. The following morning, membranes were washed three times for 5 min with TBS + 0.05% Tween-20, probed with fluorescent secondary antibodies (goat anti-rabbit IRDye 800CW and goat anti-mouse IRDye 680RD, Li-Cor) for 1 h at room temperature in the dark at 1:10,000 dilution, and washed again 3 times. Blots were imaged with the Li-Cor Odyssey CLx and analyzed using Image Studio Version 5.2.

### Antibodies
Primary antibodies used: rabbit anti-CK1α, ab206652, clone EPR19824, Abcam rabbit anti-p21, 2947S, clone 12D1, Cell Signaling Technology rabbit anti-Helios (IKZF2), 42427, clone D8W4X, Cell Signaling Technology mouse anti-beta actin, 3700S, clone 8H10D10, Cell Signaling Technology

### shRNA
MOLM-13 cells were transduced with shRNA lentiviral particles from non-targeting control (NTC) and 3 CK1α shRNAs (CSNK1a1-a, CSNK1a1-b, CSNK1a1-c; Table 2). Cells were then sorted after 3 days of culture for GFP+ cells. After sorting, 5.0E + 03 cells from each condition were harvested for immunoblot assay. An equal number of cells (5.0E + 06) from each condition were then plated in triplicates. Cells were then grown for 13 days and change in GFP+ cells was assessed by flow cytometry at days 0, 3, 6, 10, and 13 post-sort.

### MOLM-13 CRBN knock-out
CRBN⁻/⁻ MOLM-13 cells were created using CRISPR-Cas9 technology. Briefly, 1 million MOLM-13 cells were transiently transfected with pre-complexed ribonuclear proteins (RNPs) consisting of 400 pmol of chemically modified sgRNA (Synthego), 135 pmol of SpCas9 protein (St. Jude Protein Production Core), and 500 ng of pMaxGFP (Lonza) via nucleofection (Lonza, 4D-Nucleofector™ X-unit) using solution SF and program EO100 in a 100 µl cuvette according to the manufacturer's recommended protocol. Transfected cells (GFP+) were single-cell sorted by flow cytometry (St. Jude Flow Cytometry and Cell Sorting Shared Resource) into 96-well tissue culture-treated plates 5 days post nucleofection. Cells were clonally expanded and screened for the desired targeted modification (out-of-frame indels) via targeted deep sequencing using gene-specific primers with partial Illumina adapter overhangs as previously described[58]. Genotyping of clones was performed using CRIS.py[59]. Knockout clones were identified as clones containing only out-of-frame indels. PowerPlex® Fusion System

**Table 2 | shRNA plasmid information**

| Name | Company | Catalog # | Target sequence |
|---|---|---|---|
| NTC shRNA | GeneCopoeia | CSHCTR001-1-LVRU6GP | GCTTCGCGCCGTAGTCTTA |
| CK1α-shRNA #1 | GeneCopoeia | HSH059457-LVRU6GP-a | GGTTGTGGACAACCATTTACT |
| CK1α-shRNA #2 | GeneCopoeia | HSH059457-LVRU6GP-b | GCAGAATTTGCGATGTACTTA |
| CK1α-shRNA #3 | GeneCopoeia | HSH059457-LVRU6GP-c | CCCTGAACCATCAATATGACT |

**Table 3 | *CSNK1A1*sgRNA design**

| Name | Sequence (5'–3') |
|---|---|
| SM148.CRBN.g3 spacer | UGUAUGUGAUGUCGGCAGAC |
| CAGE1215.CSNK1A1.g1 spacer | UUCUAGUCGCCGAGAUGACA |
| CAGE1215.CSNK1A1.g7 spacer | UUUACCUUUAGCCCUUGCCA |
| Forward indexing primer (5'–3') | AATGATACGGCGACCACCGAGATCTACAC(6 bp index)ACACTCTTTCCCTACACGACGCTCTTC |
| Reverse indexing primer (5'–3') | CAAGCAGAAGACGGCATACGAGAT(10 bp index)GTGACTGGAGTTCAGACGTGTGCTC |
| SM148.hCRBN.DS.F2 | ctacacgacgctcttccgatctGCAGAGAGTGAGGAAGAAGATGA |
| SM148.hCRBN.DS.R2 | cagacgtgtgctcttccgatctGCCCATGTCCTCATCCACAA |
| CAGE1215.CSNK1A1.F | ctacacgacgctcttccgatctCACCAAATAGTGTTCCCTCCTCA |
| CAGE1215.CSNK1A1.R | cagacgtgtgctcttccgatctTGGGTACCAGCTTACTGTCTCT |

(Promega) was used to confirm final clone identification (Hartwell Center for Biotechnology at St. Jude). All clones tested negative for mycoplasma by the MycoAlert™Plus Mycoplasma Detection Kit (Lonza). CRBN knockout confirmed by immunoblot (Supplementary Material).

## Dependency assay
Dependency assay for *CSNK1A1* was performed by nucleofecting precomplexed RNP (150 pmol *CSNK1A1* gRNA with 50 pmol Cas9 protein) in 20 µl P3 solution with program EO100. Two different gRNAs were tested separately (CAGE1215.CSNK1A1.g1 and CAGE1215.CSNK1A1.g7). Following nucleofection, cells were cultured at 37 °C, 5% $CO_2$, and gDNA samples were taken at days 3, 7, 14, and 21 post transfection. Indel profiles were analyzed using CRIS.py. The sequences of the *CSNK1A1*sgRNA spacers and associated genotyping primers are listed in Table 3.

## CRBN fluorescence polarization assay
In this competitive fluorescent polarization assay Cy5 conjugated lenalidomide analog (Cy5-O-Len)[60] was used as the fluorescent probe. The assay cocktail was prepared by combining 6XHis-CRBN-DDB1 protein (200 nM) and Cy5-O-Len probe (30 nM) in 20 mM HEPES pH 7, 150 mM NaCl, 0.005% Tween-20 assay buffer. Compounds were transferred to a Corning 3821 black 384-well assay plate from a dose–response plate using a Labcyte Echo 650 Acoustic Liquid Handler (Beckman Coulter, USA). Then 20 µL of the assay cocktail was dispensed to wells of the assay plate with a Multidrop Combi Reagent Dispenser (Thermo Fisher Scientific, USA). The plates were incubated in the dark for 1 h at room temperature and then read on an Envision plate reader (Perkin Elmer, USA). Raw intensity values for each compound at each concentration were normalized to obtain % activity using the following equation: 100×[(mean(negctrl)−compound)/(mean (negctrl)−mean (posctrl))], and then pooled from replicate experiments prior to fitting. Here, negctrl and posctrl refer to the negative (DMSO) and positive controls (CC-220) on each plate. Dose-response curves were fit to the Hill equation using the GraphPad Prism (version 10). Data represent the mean of three independent determinations.

## HiBiT-tagged cell culture
CSNK1A1 (CK1α)-HiBiT HEK293 LgBiT stable CRISPR edited clonal cells (Promega, Cat. #CS3023104), HiBiT-CTNNB1 (β-catenin) HEK293 LgBiT stable CRISPR edited clonal cells (Promega, Cat. #CS302340), and HEK293 cells (ATCC, Cat. #CRL-1573) were maintained in DMEM (Gibco, Cat. #11995065) supplemented with 10% FBS (VWR, Cat.# 89510-194). IKZF1-HiBiT Jurkat LgBiT stable cells (Promega, Cat. #CS3023170) were maintained in RPMI-1640 (Gibco, Cat. #11875093) supplemented with 10% FBS.

## Degradation and cell viability assays
White 96-well assay plates (Corning, Cat. # 3017) were seeded with 30,000 cells/well in growth media for CK1α-HiBiT HEK293 LgBiT stable and HiBiT-β-catenin HEK293 LgBiT stable and incubated overnight at 37 °C and 5% $CO_2$. After the overnight incubation, the medium on the CK1α-HiBiT HEK293 LgBiT stable and HiBiT-β-catenin HEK293 LgBiT stable cells was aspirated off and replaced with $CO_2$ independent medium (Gibco, Cat. #18045088) supplemented with 10% FBS and 1x Endurazine (Promega, Cat. #N2571) substrate. IKZF1-HiBiT Jurkat LgBiT stable cells were seeded at 60,000 cells/well in 50% growth media/50% $CO_2$ independent medium supplemented with 10% FBS and 1× Endurazine substrate. Luminescence was allowed to equilibrate for all plates for 2.5 h. A dose–response curve was generated by performing 3-fold dilutions of each compound including a DMSO-only control in growth media. For live cell kinetic degradation assays, diluted compounds were added to CK1α and β-catenin plates and placed in a GloMax Discover (Promega, Cat. #GM3000) preheated to 37 °C or added to IKZF1 plates and placed in a CLARIOstar Plus (BMG LABTECH) preheated to 37 °C. Luminescence was read every 5 min for 24 h. At the end of all 24-h cycles, CellTiter Glo 2.0 (Promega, Cat. #G9242) was added to the plates, incubated for at least 10 min and luminescence was read using the GloMax Discover.

Relative protein level was determined based on normalization to the DMSO control and reported at Fractional RLUs. Degradation rate and degradation maximums were calculated from Fractional RLUs as previously described[38] using GraphPad Prism software. β-catenin stabilization time of onset for each concentration curve was calculated in GraphPad Prism by fitting an unconstrained plateau followed by one phase association model using the equation below ($Y = \text{IF}(X < X0, Y0, Y0+(\text{Plateau}−Y0)*(1−\exp(−K*(X−X0))))$). The parameter, $X0$ was used as time of onset for each compound concentration.

## Live cell ternary complex nanoBRET assays
CSNK1A1CK1α-HiBiT HEK293 LgBiT stable cells weretransfected with a HaloTag-CRBN vector (Promega, Cat. #N269A) using Fugene HD

(Promega, Cat. #E2311). Following overnight incubation, cells were replated in OptiMEM I Reduced Serum Medium, no phenol red (Gibco, Cat. #11058021) supplemented with 4% FBS, and HaloTag 618 ligand (Promega, Cat. #G9801). Plates were incubated overnight at 37 °C with 5% $CO_2$. On day 3, a dose–response curve was generated by performing 3-fold dilutions of each compound including a DMSO-only control in OptiMEM I Reduced Serum Medium, no phenol red supplemented with 4% FBS. Cells were treated with MG132 (Selleckchem, Cat. #S2619) for 30 min before being treated with the dilution series of compounds. Plates were incubated for 2 h at 37 °C with 5% $CO_2$. After 2 h, plates were read using NanoBRET Nano-Glo substrate (Promega #N1573) on a GloMax Discover following the manufacturer's instructions. No ligand controls were subtracted from each sample and normalized to the no compound DMSO control to obtain Fractional BRET values. $EC_{50}$ values were calculated for each compound curve.

## Annexin-V staining for apoptosis

Wild type and $CRBN^{-/-}$ MOLM-13 cells were grown in 12-well plates (TPP, Techno Plastic Products AG) with 500,000 cells per well in 2 ml total media volume. Cells were treated with 1 μl/ml DMSO or 1 μM compound in triplicate, and incubated for 24 or 72 h at 37 °C. To prepare samples, this buffer was used: 10 mM HEPES, 0.14 M NaCl, 2.5 mM $CaCl_2$ pH 7.4, 2% v/v FBS in UltraPure water (Invitrogen). After incubation, samples were collected in 5-ml tubes for flow cytometry (Falcon). Single-stain controls consisted of ~150 μl of multiple samples combined. Wells were washed with 1 ml of buffer that was added to the corresponding sample tube. Cells were centrifuged at 500×$g$ for 5 min at 4 °C to form a pellet. Supernatant was decanted and cells were washed again with 3 ml of buffer, then centrifuged. After decanting the supernatant, 3 μl of APC-conjugated anti-Annexin-V antibody (BioLegend) was added to each tube in the residual buffer left behind. Samples were mixed by vortex and incubated at room temperature in the dark for 20 min. Samples were then washed and centrifuged a final time, decanting the supernatant and blotting the tubes to remove residual buffer. 50 μl of buffer with DAPI was added to all appropriate samples and mixed, then analyzed by flow cytometry. Collected data was analyzed using FlowJo version 10.9.

## CK1α mutant rescue in MOLM-13 cells

Plasmids pLC-Flag-CSNK1A1 WT-Puro (#123319) and pLC-Flag-CSNK1A1 G40N-Puro (#123320) from Addgene.org (gifted by Eva Gottwein)[41] were used to make lentivirus in HEK 293T cells. 48 h after HEK 293T cells were transfected using Fugene HD reagent (Promega), lentiviral media was collected, passed through a 40-micron filter, and added to 2 million MOLM-13 cells with 1 μg/ml Polybrene transfection reagent (Millipore). MOLM-13 cells were transduced by spinfection: centrifuged for 1.5 h at 30 °C, 2000 × $g$. The media was changed after cells rested in the 37 °C incubator overnight. 48 h after spinfection, positively transduced cells were selected with 6 μg/ml puromycin. CK1α overexpression was confirmed by western blot. As described above in the Molecular Glue Library Screening section, transduced MOLM-13 cells were seeded in 384-well plates at 1250 cells per well and allowed to rest overnight. 30 nl of CK1α degraders of interest were deposited on the cells using an Echo 650 acoustic liquid handler (PerkinElmer). After 72 h of incubation at 37 °C, cell viability was measured by CTG assay. This was done twice for a total of six technical replicates.

## Oncolines® cancer cell lines

Cell lines were purchased from the American Type Culture Collection (ATCC) (Manassas, VA, USA) or the German Collection of Microorganisms and Cell Cultures (DSMZ) (Braunschweig, Germany). All experiments were carried out within ten passages of the original vials. Authenticity of the cell lines was confirmed by short tandem repeat analysis at the respective provider.

## Oncolines® SJ3149 cell viability assays

Cell viability assays were performed as previously described[61]. In brief, cells were seeded in a 384-well plate at an optimized density. The starting cell number was determined after 24 h incubation by adding ATPlite 1Step (Perkin Elmer, Groningen, the Netherlands) to each well and recording luminescence on an Envision multimode reader (Perkin Elmer, Waltham, MA). After the addition of a duplicate 9-point dilution series of SJ3149, cells were incubated for another 72 h, followed by cell counting using ATPlite 1Step. The dilution series ranged from 3.2 to 31,600 nM or was diluted 100-fold for the most sensitive cell lines to ideally capture the complete dose–response curve. Percentage viability was determined at each compound concentration by relating the observed signal to a vehicle-treated control, that is, medium containing 0.4% (v/v) DMSO. Compound $IC_{50}$ values were calculated by fitting a 4-parameter logistic curve to the percentage viability values using IDBS XLfit5 (IDBS, Guildford, UK). The $IC_{50}$ was limited to the maximum tested compound concentration when the $IC_{50}$ exceeded the maximum tested concentration. All dose–response curves were visually inspected and submitted to an $F$-test, followed by the invalidation of curves with $F > 1.5$. $^{10}\log(IC_{50}[\text{nmol/L}])$ values were used for bioinformatics analyses unless otherwise indicated.

## Oncolines® cell line genomics

Genomic status of 37 frequently altered cancer genes was retrieved from the COSMIC Cell Lines Project (v80) database[42,62]. Mutations, large deletions or amplifications, and gene translocations were considered relevant genomic alterations. Only non-synonymous alterations that directly altered the coding sequence were retained for further analysis. Furthermore, mutations were filtered for occurrence in primary patient samples. Mutation status of $TP53$ was retrieved from the DepMap database (version 22Q4)[63]. Considering the diverse spectrum of protein function-altering $TP53$ mutations, all non-synonymous mutations were retained for this gene[64]. All 38 genes were altered in at least three cell lines. For each gene, the difference in average $^{10}\log IC_{50}$ between altered and wild-type cell lines was determined. Significance of the $IC_{50}$ differences was determined by Type II analysis of variance (ANOVA). ANOVA $p$-values were subjected to Benjamini–Hochberg multiple testing correction, and associations with false discovery rate <20% were considered significant.

## Cell line TP53 missense mutation status

$TP53$ missense mutations were retrieved from the DepMap database (version 22Q4). Only cell lines which had at least one missense mutation were selected for further analysis. Cell lines were grouped based on the specific $TP53$ residue position which was affected by the missense mutation. Missense mutation positions affected in at least three cell lines were selected for further analysis. Significance of the $IC_{50}$ difference was determined as described above.

## Correlation analysis of 120 reference anti-cancer agents

$IC_{50}$ values for 120 anti-cancer reference agents profiled on 102 of the 115 cell lines were determined in a similar fashion as SJ3149 (Uitdehaag et al.)[65]. Pearson correlations were calculated between SJ3149 $^{10}\log IC_{50}$ values and those of the 120 reference agents, to determine the similarity of compound $IC_{50}$ fingerprints.

## Correlation analyses of gene expression or gene dependency

Cell line RNA sequencing-based basal gene expression values (log2(TPM + 1)) and CRISPR dependency scores were retrieved from the DepMap database (version 22Q2 and 22Q4, respectively). CSNK1A1 RNAi dependency scores were downloaded from DepMap (August 16, 2023). SJ3149 or Nutlin-3a cell line drug response was related to gene expression or gene dependency by calculating Pearson correlations between the $^{10}\log IC_{50}$ values and either gene expression values or gene dependency scores in the same cell lines.

## Pharmacokinetics determination of SJ3149

Healthy female CD1 mice (8–10 weeks old) weighing between 19 and 25 g were procured from Hylasco, India. Three mice were housed in each cage. Temperature and humidity were maintained at $22 \pm 3\,°C$ and 30–70%, respectively and illumination was controlled to give a sequence of 12 h light and 12 h dark cycle. Temperature and humidity were recorded by an auto-controlled data logger system. All the animals were provided a laboratory rodent diet (Altromin, Germany). Reverse osmosis water treated with ultraviolet light was provided ad libitum. Total 27 mice ($n = 9$/group) were divided into three groups as Group 1, Group 2, and Group 3 with a sparse sampling design (3 mice/time points). Animals in Group 1 were administered intravenously as slow bolus injection through the tail vein with solution formulation, animals in Group 2 were administered through the oral route with solution formulation, and animals in Group 3 were administered through the intraperitoneal route with solution formulation of SJ3149 at 3, 50 and 50 mg/kg, respectively. The formulation vehicle for IV was 5% v/v NMP, 5% v/v Solutol HS-15 and 90% v/v Normal saline, and for PO & IP was 10%v/v NMP, 10%v/v Solutol HS-15. 50%v/v PEG-400 and 30%v/v HPβCD (20%w/v) in saline. Blood samples (-60 μL) were collected under light isoflurane anesthesia (Surgivet®) from retro-orbital plexus from a set of three mice at 0.08 (IV only), 0.25, 0.5, 1, 2, 4, 6 (PO and IP only), 8, 12 and 24 h. Immediately after blood collection, plasma was harvested by centrifugation at 12,300×g, 10 min at 4 °C and samples were stored at $-70 \pm 10\,°C$ until bioanalysis. Concentrations of SJ3149 in mouse plasma samples were determined by fit-for-purpose LC–MS/MS method. Non-Compartmental-Analysis tool of Phoenix WinNonlin® (Version 8.3) was used to assess the pharmacokinetic parameters. Peak plasma concentration ($C_{max}$) and time for the peak plasma concentration ($T_{max}$) were the observed values. The area under the concentration–time curve ($AUC_{last}$) was calculated by the linear trapezoidal rule. The terminal elimination rate constant, ke was determined by regression analysis of the linear terminal portion of the log plasma concentration-time curve. Clearance was estimated as Dose/$AUC_{inf}$ and Vss as CL*MRT. The oral bioavailability was calculated as the ratio of dose-normalized AUC of the oral and intravenous groups multiplied by 100.

The PK study was conducted at the AAALAC-accredited facility of Sai Life Sciences Limited, Pune, India, in accordance with the Study Protocol SAIDMPK/PK-20-08-538. All animals were maintained in standard animal cages under conventional laboratory conditions (12 h/12 h light/dark cycle, 22 °C) with ad libitum access to food and water. All procedures of the present study were performed in accordance with the guidelines provided by the Committee for the Purpose of Control and Supervision of Experiments on Animals (CPCSEA) as published in The Gazette of India, on December 15, 1998. Prior approval from the Institutional Animal Ethics Committee (IAEC) was obtained before the initiation of the study.

## In vivo MOLM-13 xenograft and SJ3149 treatment

Female NSG mice (NOD.Cg-*Prkdc*[scid] *Il2rg*[tm1Wjl]/SzJ, Strain #005557, The Jackson Laboratory) aged 6–8 weeks were transplanted with 100,000 MOLM-13 cells by tail-vein injection. 10 days post-transplantation, mice were treated with SJ3149 via intraperitoneal injections for the following 2-day dosing schedule: 50 mg/kg (1 dose every 12 h), and 50 mg/kg (1 dose every 24 h) with $n = 3$ per group. Control groups ($n = 3$) were treated with DMSO either 1 dose every 24 h or 1 dose every 12 h with equal volume as treatment groups. Mice were euthanized 48 h after the first treatment and bone marrow cells were collected. Murine cells were depleted using the EasySep Mouse/Human Chimera isolation kit (STEMCELL Technologies #19849). NSG mice were housed in micro-isolation cages and kept under barrier conditions to keep them specific pathogen-free. The lights for NSG mice are on an automated 12-h on, 12-h off light cycle. The mouse room has a separate thermostat and humidistat to control temperature and humidity at the room level. Sex was not considered beyond female mice being most commonly used for cell line-derived xenografts. Ethical approval by the St. Jude Institutional Animal Care and Use Committee.

## Protein expression and purification

Truncated human CRBN (StrepII-CRBN[Δ1-40]) and human DDB1 (His$_6$-DDB1[ΔBPB])[66] were co-expressed in *Trichoplusia ni* High-Five insect cells using the BestBac 2.0 baculovirus expression system (Expression Systems). Human full-length CK1α (StrepII-CK1α) was synthesized and cloned into the pFastBac1 vector (Genscript) and expressed in High-Five insect cells using the Bac-to-Bac baculovirus expression system (Invitrogen). StrepII-CRBN[Δ1-40]-His$_6$-DDB1[ΔBPB] cells were resuspended in 50 mM Tris–HCl pH 8.0, 200 mM NaCl, 1 mM TCEP, 10% glycerol, 50 μM Zinc Acetate, and SigmaFast protease cocktail inhibitor (Millipore Sigma). StrepII-CK1α cells were resuspended in 50 mM Tris–HCl pH 8.0, 500 mM NaCl, 0.25 mM TCEP, 1 mM PMSF, 10% glycerol, and SigmaFast protease cocktail inhibitor. Both StrepII-CRBN[Δ1-40]-His$_6$-DDB1[ΔBPB] and StrepII-CK1α were lysed by sonication (2 s ON, 2 s OFF, 3 min) and clarified by centrifugation (136,000×g) for 1.5 h at 4 °C and loaded onto StrepTactin Sepharose resin (IBA) equilibrated in 50 mM Tris–HCl pH 8.0, 200 mM NaCl, 0.25 mM TCEP, and 10% glycerol. Resin was washed and proteins were eluted with equilibration buffer supplemented with 50 mM Biotin. The StrepII fusion tag of CK1α was cleaved using TEV protease (PPF). Both cleaved CK1α and StrepII-CRBN[Δ1-40]-His$_6$-DDB1[ΔBPB] were buffer exchanged and purified by size-exclusion chromatography (S200 10/300 Increase) in the presence of 50 mM HEPES pH 7.4, 200 mM NaCl, and 0.25 mM TCEP. Protein fractions were pooled and concentrated to 70 μM (StrepII-CRBN[Δ1-40]-His$_6$-DDB1[ΔBPB]) and 80 μM (CK1α) for crystallization experiments.

## Crystallization and data collection

For crystallization of the StrepII-CRBN[Δ1-40]-His$_6$-DDB1[ΔBPB]-SJ3149-CK1α complex, 70 μM StrepII-CRBN[Δ1-40]-His$_6$-DDB1[ΔBPB] was mixed and incubated with 35 μM SJ3149 before the addition of 80 μM CK1α. The mixture was incubated on ice for 1 h and subsequently centrifuged at 20,000×g for 30 min at 4 °C. 24-well sitting drop crystallization plates (Hampton) were set up by mixing protein 1:1 with reservoir solution containing 70 mM Tris pH 7.0, 140 mM MgCl$_2$ and 10% (w/v) PEG8000 and plates were incubated at room temperature. Crystals appeared after 24 h and continued growing for 14 days via vapor diffusion. Crystals were cryo-protected in a reservoir solution supplemented with 20% ethylene glycol and flash-cooled in liquid nitrogen. Diffraction data were collected at SER-CAT (beamline 22-ID) to a resolution of 3.45 Å.

## Structure determination and model building

Data collection and refinement statistics are provided in Table SB1. The P1 data was integrated and scaled in XDS, scaling two different datasets from the same crystal using XSCALE[67]. The structure was solved by Molecular Replacement with 5FQD as the search model using Phaser[68]. The asymmetric unit contained two ternary complexes, with ABC and DEF corresponding to DDB1, CRBN, and CK1alpha, respectively. SJ3149 is unambiguously modeled at the CRBN−CK1 alpha interface in the ABC quaternary complex. SJ3149 was not modeled in the DEF complex due to insufficient electron density. Model building and refinement were performed in COOT[69] and Phenix[70], respectively. The ABC complex electron density is of higher quality relative to DEF and is referred to throughout the manuscript.

## Reporting summary

Further information on research design is available in the Nature Portfolio Reporting Summary linked to this article.

# Data availability

The proteomics data generated in this study are available from the ProteomeXchange Consortium via the PRIDE partner repository under

accession codes PXD045170, PXD045228, and PXD045230, corresponding to the experiments related to SJ7095, SJ0040, and SJ3149, respectively. The crystallographic data for the quaternary complex of CK1α + CRBN$^{\Delta 1-40}$ + DDB1$^{\Delta BPB}$ + SJ3149 have been deposited into the Protein Data Bank under accession codes 8G66 [https://doi.org/10.2210/pdb8G66/pdb]. All other data supporting the findings of this study are available within the article and its supplementary files. Any additional requests for information can be directed to the corresponding authors. Source data are provided as a Source Data file with this paper. Source data are provided with this paper.

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

## Acknowledgements

We are grateful for the support of the American Lebanese Syrian Associated Charities (ALSAC) and St. Jude Children's Research Hospital, and we would like to thank the patients, their families, and the staff at our institution. This work was also supported in part by the Alex's Lemonade Stand Foundation Crazy 8 award. We thank A. Panchmatia for her help with the protein preparation that led to the structure, and A.S. Tanwar for the initial expression test and purification. Data were collected at the Southeast Regional Collaborative Access Team (SER-CAT) 22-ID (or 22-BM) beamline at the Advanced Photon Source, Argonne National Laboratory. SER-CAT is supported by its member institutions, and equipment grants (S10_RR25528, S10_RR028976, and S10_OD027000) from the National Institutes of Health. C.G.M. received a grant from the National Cancer Institute (R35197695). We also thank the St. Jude Biomolecular X-Ray Crystallography Center for support. We thank Analytical Technologies Center at the Department of Chemical Biology and Therapeutics for generating in vitro ADME data for SJ3149. We also acknowledge the Compound Management Center at the Department of Chemical Biology and Therapeutics for performing general QC and compound plate reformatting for screening. We thank Madison Rice for editing the figures. Use of the Advanced Photon Source was supported by the U.S. Department of Energy, Office of Science, Office of Basic Energy Sciences, under Contract No. W-31-109-Eng-38. Data were collected at the Southeast Regional Collaborative Access Team (SER-CAT) 22-ID beamline at the Advanced Photon Source, Argonne National Laboratory. SER-CAT is supported by its member institutions (see www.ser-cat.org/members.html), and equipment grants (S10_RR25528 and S10_RR028976) from the National Institutes of Health.

## Author contributions

Z.R. conceived the study. J.M.K. directed biology studies. M.F. directed structural biology efforts. G.N. directed chemistry. S.Y. purified proteins; S.M.Y. and D.M. generated the crystal structure. S.A. and L.G.M. generated immunoblots, shRNA, CRISPR, and annexin data. S.D. and A.A.S. conducted in silico modeling. A.A.H., X.W., K.Y., S.Z., and J.P. performed proteomics studies. S.N., A.J.L., and S.M.P-M. generated CRBN⁻/⁻ MOLM-13 cells. K.M. and Z.S. performed chemistry synthesis. C.G.M., L.G.M., J.M.B., R.H., Y.C., V.M., A.A., and A.M. performed biochemical and cell-based assay screens. R.K. provided pellets for purification. E.A.C., D.L.D., M.U., and K.M.R. generated degradation

kinetics data. J.J.K. and G.J.R.Z. performed Oncoline data analysis. M.T. carried out in vivo PD experiment. The paper was written by G.N. and Z.R. with input from all authors.

## Competing interests

The authors declare the following competing interests: St. Jude Children's Research Hospital has applied for an international patent covering structures reported in this work (WO2023081224A1; and provisional application number: 63389477), G.N., K.M., J.M.K., and Z.R. The remaining authors declare no competing interests.

## Additional information

[1]Department of Chemical Biology and Therapeutics, St. Jude Children's Research Hospital, 262 Danny Thomas Place, Memphis, TN 38105, USA. [2]Department of Pathology, St. Jude Children's Research Hospital, 262 Danny Thomas Place, Memphis, TN 38105, USA. [3]Promega Corporation, 5430 East Cheryl Drive, Madison, WI 53711, USA. [4]Oncolines B.V., Kloosterstraat 9, 5349 AB Oss, The Netherlands. [5]Department of Structural Biology, St. Jude Children's Research Hospital, 262 Danny Thomas Place, Memphis, TN 38105, USA. [6]Center for Advanced Genome Engineering, St. Jude Children's Research Hospital, 262 Danny Thomas Place, Memphis, TN 38105, USA. [7]Center for Proteomics and Metabolomics, St. Jude Children's Research Hospital, 262 Danny Thomas Place, Memphis, TN 38105, USA. [8]Department of Cell and Molecular Biology, St. Jude Children's Research Hospital, 262 Danny Thomas Place, Memphis, Memphis, TN 38105, USA. [9]These authors contributed equally: Gisele Nishiguchi, Lauren G. Mascibroda, Sarah M. Young. ✉e-mail: marcus.fischer@stjude.org; jeffery.klco@stjude.org; zoran.rankovic@stjude.org

