## [Peer Review File · Nature Communications]

Reviewers' Comments:

Reviewer #1:

Remarks to the Author:

The authors outline a screening approach towards the discovery of selective degraders of CK1a. They take a proprietary library of CRBN targeted chemical matter and screen for impact on the viability of nine human paediatric cancer cell lines, identifying an initial hit that impacts the viability only of MOLM-13 cells. Whole proteome profiling combined with analysis of publicly available cancer dependency data sets highlighted CK1a as a likely target protein for further optimisation. Optimisation is then performed and a more potent CK1a degrader probe molecule is studied for its impact across a much wider set of cancer cell lines. Quaternary co-crystal x-ray structure data is then provided and analysed to enable hypotheses around why this molecule shows enhanced CK1a degradation compared with lenalidomide.

This study provides a new best in class selective CK1a degrader that is more potent and/or selective than those previously published by other groups. This in turn enables illumination of CK1a cancer dependencies and proposing CK1a as a valid target for cancers containing active wild type p53.

This study will be of interest and value to those in cancer research fields and even more so to a considerable number of researchers actively studying molecular glue based drug discovery and CRBN based degrader discovery.

However, major claims of the study include 1. That they have discovered a potent and selective CK1a degrader and 2. That this can be explained by the provided structural data of the quaternary structure and was indeed achieved via rational design. I am not convinced that there is sufficient evidence provided yet to fully support those claims, but would encourage the authors to consider the following feedback towards either modifying or better supporting those claims:

1. Most critically, no data specifically for IKZF2 or IKZF4 is shown and it is not clear if IKZF2 and IKZF4 were quantified in the proteomics data set – this needs to be done to demonstrate clear selectivity versus these transcription factors, given that chemical matter not too far removed from these structures has shown dual CK1a/IKZF2 degradation but selectivity over IKZF1/3 (Woo lab reference 25). In fact, the histidine for glutamine switch in IKZF2/4 versus 1/3 may well be relevant to the binding mode the molecules from this study and that previous study by Woo et al did not pick out IKZF2 as a degradation hit so clearly in their own whole cell proteomics data but was shown via western. Confirming this via western blot, analysing endogenous IKZF2 levels in MOLM-13 cells following SJ3149 is required here to confirm selectivity and interpretation of subsequent biological data prior to publication of this study.

2. The data shown in extended figure 3 does not seem to support that knockdown of CK1a is independently responsible for impact on cell viability. The knockdown seems only partial and I could not see any accompanying viability data. A much more convincing effect is seen with the probe molecules developed in this paper, which again highlights the need to demonstrate the need to show definitively this is not due to a mixed effect of IKZF2 and CK1a degradation.

3. In the section titled 'Structure-guided design leads to the discovery of selective CK1a degraders.' The only hypothesis presented is that the authors were seeking to place chemical motifs in the C5 position towards where the beta hairpin loop of CK1a is expected to be placed. However, no other explanation is given as to why the chemical functionalities (i.e. aromatic groups) chosen were used e.g. what residues were being targeted and in what way is this validated? Without this I don't feel a claim can be made that this is a structure guided approach as it is likely that within the >3000 molecule library there are molecules that have functional groups in the C5 position anyway. I would suggest the title of this section is modified, perhaps stressing more exploration of structure activity relationships rather than rational design? Along these lines, the claims made in the introduction and abstract need to be toned down to state that the observed structure activity relationships was rationalised via computational co-crystal structure data, rather than guided by it.

4. On line 360 the authors say "residue E377 rotates to form a direct rather than a water361 bridged interaction with SJ3149 (Fig. 4d)". However, it is impossible for this to be verified/supported with the data provided. The resolution of the structure is not particularly high, but does not necessarily preclude this conclusion if there is a sufficiently large difference in the electron densities around this residue between the new structure here and the previously published lenalidomide structure. The authors need to at least show in a figure the electron density for this residue and compare it with the known structure with lenalidomide to convince that this conclusion is sound.

5. On line 396 the authors state "This stabilization of closed CRBN is thought to increase CK1a binding and enhance the stability of the ternary complex." They then use this as a basis for rationalising SJ3149's increased potency for CK1a degradation compared with lenalidomide, explaining that an extra interaction formed by SJ3149 stabilises the closed form. However, this explanation/rationale is only valid if the first statement on line 396 is true and supported by data. Therefore, I would ask that either a reference supporting the first statement on 396 is given (that is specific to CK1a binding to the closed form of CRBN) or data is provided to support that.

Reviewer #2:

Remarks to the Author:

In this manuscript, Nishiguchi and colleagues describe the identification and characterization of SJ3149, a highly potent and selective CK1a degrader that features a broad efficacy in a panel of cancer cell lines.

They set out to screen an extensive chemical library of >3000 CRBN-dependent molecular glue degrader candidates via viability assays in several pediatric leukemia cell lines. SJ7095 stood out in its activity to selectively kill MOLM13 cells. Target identification was conducted via quantitative proteomics, revealing that SJ7095 potently degraded CK1a in addition to IKZF1/3. Chemical optimization afforded several analogues, including SJ3149, with increased potency and selectivity, as assessed via a comprehensive suite of assays. Next, cell line profiling of SJ3149 indicated a broad efficacy over many cancer cell lines, also beyond AML. Correlative studies with existing data suggest that p53 status as the main criterion for the observed selectivity. Finally, the authors solve the structure of the ternary complex of CRBN:DDB1, SJ3149 and CK1a, thus providing molecular insights to the observed potency and selectivity.

In sum, this is an excellent study that is well executed and written. While CK1a degraders have been previously published, this study stands out with regards to the quality of the presented research. In principle, I strongly recommend publication in Nature Communications. There is one experiment that I think would nicely connect all datasets and that I'd highly recommend to run prior to publishing (see major point below). All other points I list below are minor points.

Major points:

1. While all data is coherent with CK1a being the main driver of efficacy, a rescue experiment would be the ultimate proof of this important thesis. I would recommend that the authors stably & ectopically overexpress a CK1a cDNA where the central glycine on the G-loop is mutated. Assuming the mutation is functional, the logic would be that this mutation renders CK1a insensitive to SJ3149 (no degradation observed). As a consequence, one would expect a strong shift in cellular sensitivity to SJ3149.
2. Figure 3K: I don't understand why CK1a is not among the top hits. Assuming that SJ3149 is primarily toxic due to CK1a ablation, then one would expect that CK1a knockout is the best-correlated perturbation over the assayed cell line panel. The authors should please explain.

Minor points:

1. Personally, I would refer to the structure as "ternary structure".
2. When citing the Ebert paper (Kroenke et al, ref 2), the authors should also refer to the back to back paper from the Kaelin group
3. Fig 3C is somewhat unclear to me and would benefit from a better description in the text or Figure legend.
4. Fig 3K: I would remove the speculation around CRBN. It is a "never essential" gene and, in my opinion, unlikely to contribute to the efficacy of SJ3149.

Reviewer #3:

Remarks to the Author:

Nishiguchi, Adelhamed, Young, and colleagues describe an exciting approach to identify and optimize molecular glues targeting CK1a. Following a library screen, the authors developed SJ3149, which displays selectivity towards CK1a. The degraders are well-characterized and meet the standards in the TPD field, with structural support to guide the rationalization of the improved effects. The applications in hematological and solid tumor cancers are also interesting. I think that this manuscript is clear and well-written and will be of wide interest to the TPD community. In advance of potential publication in Nature Communications, I have a few concerns for the authors to address.

Major points

- 1) The implications for in vivo evaluation are unclear and suitability for in vivo studies should be established. In vitro ADME data and pharmacokinetic studies are needed to support the potential of their compounds for in vivo evaluation.
- 2) The authors noted in line 347 that dependency on CRBN is correlative with responses. Is this also related to CRBN expression levels?
- 3) The reported connection to P53 activity is interesting and seems to support the prior publication that the authors cited. However, the proteomics in Fig. 2j lacks the time-dependent evaluation to show that CK1a degradation then drives the response. The authors should perform time courses (e.g. western blots) to help support that CK1a degradation occurs prior to the stabilization of P53, P21, and B-catenin. Are these responses also related to E2F1, which is involved in the model that the authors cited in Huart et al?
- 4) The authors should discuss the potential limitations or challenges of their screening approach in the discussion.

Minor points

- 1) The reference numbering should be double-checked. For example, in line 83, reference 26 seems like it should be reference 27.
- 2) The heat map in Fig. 1a was difficult to interpret. Can the compound IDs be listed? In addition, can the quality of Fig. 1c be improved? The colors made it difficult to interpret.
- 3) I appreciate the data in Fig. 2. Can a summary table be provided to help orient the reader in IC50s, Dmax, etc that are noted in the text?
- 4) For several legends, key information is lacking to help interpret the data. For example, how many samples are represented, and are they biological/technical replicates?

October 27, 2023

Dear Reviewers,

Thank you all for the insightful and thoughtful comments. We have addressed all comments and revised the manuscript accordingly. Below is a list of point-by-point responses and the corresponding changes that we made.

Yours sincerely,

Zoran Rankovic, for the authors.

RESPONSE TO REVIEWER COMMENTS

Reviewer #1

1. Most critically, no data specifically for IKZF2 or IKZF4 is shown and it is not clear if IKZF2 and IKZF4 were quantified in the proteomics data set – this needs to be done to demonstrate clear selectivity versus these transcription factors, given that chemical matter not too far removed from these structures has shown dual CK1a/IKZF2 degradation but selectivity over IKZF1/3 (Woo lab reference 25). In fact, the histidine for glutamine switch in IKZF2/4 versus 1/3 may well be relevant to the binding mode the molecules from this study and that previous study by Woo et al did not pick out IKZF2 as a degradation hit so clearly in their own whole cell proteomics data but was shown via western. Confirming this via western blot, analysing endogenous IKZF2 levels in MOLM-13 cells following SJ3149 is required here to confirm selectivity and interpretation of subsequent biological data prior to publication of this study.

Response: This is an important point raised by Reviewer #1, especially when considering the compound described by Woo and colleagues. According to the TMT proteomics analysis, there was a negligible change in IKZF2 protein levels after treatment, and IKZF4 was not detected. To formally address this question MOLM-13 cells were treated with SJ3149 for 4 hours at

concentrations ranging from 1 nM to 10 μ M and analyzed by western blot to quantify IKZF2 protein levels. The combined results from 3 replicates show that SJ3149 does lead to some reduction in IKZF2 expression, but to a significantly lesser extent than CK1a. While CK1a is 50% degraded at 4 nM SJ3149, IKZF2 is reduced by only ~40% at 10 μ M dosing for 4 hours. These new data are included in the manuscript (Extended Data Fig. 6d). At low concentrations that can be used to target CK1 α there is little effect on IKZF2 expression levels further suggesting that SJ3149 has selectivity for CK1 α over IKZF2.

2. The data shown in extended figure 3 does not seem to support that knockdown of CK1a is independently responsible for impact on cell viability. The knockdown seems only partial and I could not see any accompanying viability data. A much more convincing effect is seen with the probe molecules developed in this paper, which again highlights the need to demonstrate the need to show definitively this is not due to a mixed effect of IKZF2 and CK1a degradation.

Response: We appreciate this point regarding the knockdown data. To further support our findings we evaluated the available Cancer Dependency Map CRISPR data for MOLM-13 cells, which showed that MOLM-13 cells are dependent on CK1 α but not on any of the IKZF proteins. This in combination with the CK1 α /IKZF2 western blot data discussed above (Extended Data Fig. 6d), as well as the low endogenous expression level of IKZF2 in MOLM-13 cells, makes us confident that the primary cell viability effects of SJ3149 that we see are due to loss of CK1 α .

3. In the section titled ‘Structure-guided design leads to the discovery of selective CK1 α degraders.’ The only hypothesis presented is that the authors were seeking to place chemical motifs in the C5 position towards where the beta hairpin loop of CK1a is expected to be placed. However, no other explanation is given as to why the chemical functionalities (i.e. aromatic groups) chosen were used e.g. what residues were being targeted and in what way is this validated? Without this I don’t feel a claim can be made that this is a structure guided approach as it is likely that within the >3000 molecule library there are molecules that have functional groups in the C5 position anyway. I would suggest the title of this section is modified, perhaps stressing more exploration of structure activity relationships rather than rational design? Along these lines, the claims made in the introduction and abstract need to be toned down to state that the observed structure activity relationships was rationalised via computational co-crystal structure data, rather than guided by it.

Response: As the reviewer suggested, we replaced throughout the manuscript the “structure-guided design” term with “structure-guided SAR exploration”. In addition, please note that while indeed there were many compounds in our library containing C5-substitution, none of them showed activity in the initial screen. It was only when guided by the in silico model of the hit compound, SJ7095, that we synthesized C5-derivatives, which ultimately led to identification of SJ0040 with a high potency in this cell line.

4. On line 360 the authors say ‘‘residue E377 rotates to form a direct rather than a water361 bridged interaction with SJ3149 (Fig. 4d).’’. However, it is impossible for this to be verified/supported with the data provided. The resolution of the structure is not particularly high, but does not necessarily preclude this conclusion if there is a sufficiently large difference in the electron densities around this residue between the new structure here and the previously published lenalidomide structure. The authors need to at least show in a figure the electron density for this residue and compare it with the known structure with lenalidomide to convince that this conclusion is sound.

Response: Excellent point – we replaced “except that residue E377 rotates to form a direct rather than a water bridged interaction with SJ3149 (Fig. 4d)” with “including a direct interaction between residue E377”. We also included the electron density evidence for the direct interaction between E377 and SJ3149 (rendered at 0.82 sigma) – Extended Data Fig. 8a.

5. On line 396 the authors state ‘‘This stabilization of closed CRBN is thought to increase CK1 α binding and enhance the stability of the ternary complex.’’ They then use this as a basis for rationalising SJ3149’s increased potency for CK1 α degradation compared with lenalidomide, explaining that an extra interaction formed by SJ3149 stabilises the closed form. However, this explanation/rationale is only valid if the first statement on line 396 is true and supported by data. Therefore, I would ask that either a reference supporting the first statement on 396 is given (that is specific to CK1 α binding to the closed form of CRBN) or data is provided to support that.

Response: To address this concern we deleted the above-mentioned sentence: ‘‘*This stabilization of closed CRBN is thought to increase CK1 α binding and enhance the stability of the ternary complex.*’’ and modified the following sentence to rationalize the improved degradation profile of SJ3149: ‘‘*With SJ3149 forming a hydrogen bond with K18 and additional hydrophobic contacts with the CK1 α degron-loop (Fig. 4b, d) that is absent in the lenalidomide-bound complex,*

our structure proposes a molecular basis for SJ3149's increased ternary complex stabilization and consequent CK1 α degradation potency compared to lenalidomide."

Reviewer #2

Major points:

1. While all data is coherent with CK1 α being the main driver of efficacy, a rescue experiment would be the ultimate proof of this important thesis. I would recommend that the authors stably & ectopically overexpress a CK1 α cDNA where the central glycine on the G-loop is mutated. Assuming the mutation is functional, the logic would be that this mutation renders CK1 α insensitive to SJ3149 (no degradation observed). As a consequence, one would expect a strong shift in cellular sensitivity to SJ3149.

Response: This is great suggestion by Reviewer #2 to further confirm that CK1 α is the target for SJ3149. We obtained FLAG-tagged CK1 α with the G40N G-loop mutation, along with a wild-type CK1 α control in the same viral backbone, and transduced MOLM-13 cells. After puromycin selection, these MOLM-13 cells overexpressing either wild type FLAG-CK1 α or the G40N mutant were treated in triplicate with a dilution series of SJ3149 and cell viability was analyzed by CTG assay. The cells expressing the G40N protein are no longer sensitive to SJ3149, demonstrating that the mutant protein rescues the cell viability phenotype. The effect of SJ3149 on the expression of CK1 α in these cells was also visualized by western blot. Both endogenous CK1 α and the overexpressed wild-type CK1 α are degraded while the G40N mutant protein is not, which supports the CTG data. These new data are included in Extended Data Fig.7.

2. Figure 3K: I don't understand why CK1 α is not among the top hits. Assuming that SJ3149 is primarily toxic due to CK1 α ablation, then one would expect that CK1 α knockout is the best-correlated perturbation over the assayed cell line panel. The authors should please explain.

Response: This is most likely because CK1 α (CSNK1A1 gene) is a common essential gene, when knocked out using CRISPR. This explanation is supported when the correlation of cell line response to SJ3149 is compared between CRISPR and RNAi, where CRISPR results in knockout and RNAi only in downregulation (knockdown). Thus, while CSNK1A1 CRISPR knock out is not significantly correlated ($r = 0.18$, $p = 0.12$), the correlation is improved and significant when

SJ3149 cell line responses are correlated with CSNK1A1 RNAi values ($r = 0.26$; $p = 0.03$) (<https://depmap.org/portal/gene/CSNK1A1?tab=overview>, accessed July 27th, 2023).

To clarify this point, we included the following statement in the manuscript: “*Notably, CSNK1A1 CRISPR dependency scores were not significantly correlated with response to SJ3149 ($r = 0.18$; p -value = 0.12). Interestingly, CSNK1A1 was previously identified as a pan-essential gene in two distinct CRISPR screens.⁴⁸ It has been proposed that, for CRISPR pan-essential genes, knockdown using RNAi might better reflect pharmacologic inhibition of the protein target, owing to the partial gene suppression induced by RNAi.⁴⁹ Indeed, the correlation between SJ3149 IC50 values and CSNK1A1 RNAi scores exhibited a statistically significant improvement ($r = 0.26$; p -value = 0.03).*”

Minor points:

1. Personally, I would refer to the structure as “ternary structure”.

Response: As this is a direct comparison to 5FQD, where the lenalidomide-bound CK1 α +CRBN+DDB1 structure is termed “quaternary”, we will continue the original terminology to avoid confusion. While reviewing this comment, we also corrected sub-section title from “*Structural basis of CK1 α +CRBN+DDB1 ternary complex formation by SJ3149*” to “*Structural basis of CK1 α +CRBN+DDB1 quaternary complex formation by SJ3149*”.

2. When citing the Ebert paper (Kroenke et al, ref 2), the authors should also refer to the back to back paper from the Kaelin group.

Response: Great point, thanks! We now cite Kaelin’s paper (Ref# 3).

3. Fig 3C is somewhat unclear to me and would benefit from a better description in the text or Figure legend.

Response: We have modified the legend for 3c to: “*SJ3149 IC50 values for the 115 cancer cell lines relative to the panel average IC50. A negative value indicates a below average IC50 value.*”

4. Fig 3K: I would remove the speculation around CRBN. It is a “never essential” gene and, in my opinion, unlikely to contribute to the efficacy of SJ3149.

Response: We have removed this statement, as suggested (lines 346-349).

Reviewer #3

Major points

1. The implications for in vivo evaluation are unclear and suitability for in vivo studies should be established. In vitro ADME data and pharmacokinetic studies are needed to support the potential of their compounds for in vivo evaluation.

Response: We agree, this is indeed an important point. We have generated PK data for our lead compound SJ3149 (Extended Data Fig. 11a,b). Based on the findings from the PK studies we also performed a PD study in which NSG mice were engrafted with MOLM13 cells and treated with three different doses of SJ3149 over two days, to determine this compound's effect on CK1a protein levels in vivo (Extended Data Fig. 13a,b).

2. The authors noted in line 347 that dependency on CRBN is correlative with responses. Is this also related to CRBN expression levels?

Response: This is an interesting question. It appears that the response is not related to CRBN expression levels. The correlation between SJ3149 IC₅₀ values and CRBN gene expression was -0.13 (p = 0.2). Upon request of reviewer #2, the sentence about CRBN dependency and response to SJ3149 has been removed from the text (lines 346-349) and Fig. 3K.

3. The reported connection to P53 activity is interesting and seems to support the prior publication that the authors cited. However, the proteomics in Fig. 2j lacks the time-dependent evaluation to show that CK1a degradation then drives the response. The authors should perform time courses (e.g. western blots) to help support that CK1a degradation occurs prior to the stabilization of P53, P21, and B-catenin. Are these responses also related to E2F1, which is involved in the model that the authors cited in Huart et al?

Response: We agree that a time course experiment would be a valuable experiment to further confirm that CK1 α degradation is the main driver of the observed cellular phenotypes. To this end, we performed an 8-hour time course on MOLM-13 cells treated with 1 μ M SJ3149 in triplicate and analyzed p21 levels by western blot. We opted to remain focused on CK1a and p21 expression to be consistent with the original dose response blots. We see a clear and almost complete reduction in CK1 α levels after 1 hour. In contrast, the increase in p21 protein levels does not occur until after 4 hours. This delay between CK1 α degradation and the increase in p21 levels further supports our overall story. We do, however, recognize that p21 is not the only protein likely to show a similar increase with protein levels in response to CK1 α degradation and that β -

catenin and E2F1 are additional likely candidates. Of note, we also observed an increase in β -catenin protein levels (see Figure 2J) after SJ3149 treatment but no increase in E2F1 levels were detected in our proteomics data. To further support our findings on β -catenin, we have also recently observed an increase in β -catenin levels using a HiBit assay after treatment with SJ3149, which parallels our p21 findings. We have included these new data in Extended Data Fig. 6a,b.

4. The authors should discuss the potential limitations or challenges of their screening approach in the discussion.

Response: We included following paragraph in the discussion section: *“Despite these results, there are several limitations of our phenotypic screening approach that relies solely on the cell viability as the assay readout. One is that degraders of proteins that are not essential for the cells featuring in the screening panel would be missed, while they could still be oncodrivers in cancers not represented in the panel. For example, since none of the cell lines in our panel are sensitive to IKZF1 degradation we would miss discovering IKZF1 degraders, which are a component of the front-line therapies for multiple myeloma. In addition, weaker degraders exerting only a small effect on the cell viability may also be missed, even if they could provide an attractive starting point for optimization. Moreover, degradation of previously reported neosubstrates associated with a strong phenotype, such as GSPT1, can generate a lot of unproductive follow up work.”*

Minor points

1. The reference numbering should be double-checked. For example, in line 83, reference 26 seems like it should be reference 27.

Response: Corrected.

2. The heat map in Fig. 1a was difficult to interpret. Can the compound IDs be listed? In addition, can the quality of Fig. 1c be improved? The colors made it difficult to interpret.

Response: The heat map with compound numbers (many ~10 digits) looks even more confusing. We increased the font size, which seems to improve the visibility. Colors in Fig.1c were made sharper and the color of the ligand was changed to make it more visible.

3. I appreciate the data in Fig. 2. Can a summary table be provided to help orient the reader in IC50s, Dmax, etc that are noted in the text?

Response: Great suggestion - we summarized all data mentioned in the text in Extended Data Table 1.

4. For several legends, key information is lacking to help interpret the data. For example, how many samples are represented, and are they biological/technical replicates?

Response: We incorporated the requested information in all figure captions.

Reviewers' Comments:

Reviewer #1:

Remarks to the Author:

Many thanks for fully addressing my feedback. Congratulations on a great study.

Reviewer #2:

Remarks to the Author:

The authors have addressed my concerns. Of particular relevance is the very clear rescue on degradation and viability that the authors observe with the G-loop mutation in CK1a.

(as a minor comment: please check labeling of extended data fig 7. It currently states [um vs uM].)

I congratulate the authors to a very complete, well-controlled, and important study.

Reviewer #3:

Remarks to the Author:

The authors have sufficiently addressed my comments. The manuscript is suitable for publication and will be of broad interest in the field. I look forward to seeing their publication in print.

December 11, 2023

Dear Reviewers,

We would like to thank you All for your positive feedback as well as for insightful and thoughtful comments, that led to a significantly strengthened revised submission. Please see below for a list of point-by-point responses and the corresponding changes that we made.

Yours sincerely,

Zoran Rankovic, for the authors.

RESPONSE TO REVIEWER COMMENTS

Reviewer #1: Many thanks for fully addressing my feedback. Congratulations on a great study.

Response: Thank You for your comments and positive feedback.

Reviewer #2: The authors have addressed my concerns. Of particular relevance is the very clear rescue on degradation and viability that the authors observe with the G-loop mutation in CK1a. (as a minor comment: please check labeling of extended data fig 7. It currently states [μm vs μM].) I congratulate the authors to a very complete, well-controlled, and important study.

Response: Extended Data Fig 7 (now Supplementary Fig 7) is corrected as advised. Thank You for your comments and positive feedback.

Reviewer #3: The authors have sufficiently addressed my comments. The manuscript is suitable for publication and will be of broad interest in the field. I look forward to seeing their publication in print.

Response: Thank You for your comments and positive feedback.